

# Comparison of simulated and reconstructed variations in East African hydroclimate over the last millennium

François Klein[1], Hugues Goosse[1], Nicholas E. Graham[2], and Dirk Verschuren[3]

[1]Georges Lemaître Centre for Earth and Climate Research (TECLIM), Earth and Life Institute, Université catholique de Louvain (UCL), Belgium
[2]Hydrologic Research Center, San Diego, CA, USA
[3]Limnology Unit, Department of Biology, Ghent University, Belgium

*Correspondence to:* François Klein (francois.klein@uclouvain.be)

**Abstract.** The multi-decadal to centennial hydroclimate changes in East Africa over the last millennium are studied by comparing the results of forced transient simulations by six General Circulation Models (GCMs) with published hydroclimate reconstructions from four lakes: Challa and Naivasha in equatorial East Africa, and Masoko and Malawi in southeastern inter-tropical Africa. The GCMs simulate fairly well the unimodal seasonal cycle of precipitation in the Masoko/Malawi region

and the bimodal seasonal cycle characterizing the Challa/Naivasha region, except that in the latter the relative magnitude of the two rainy seasons is less well captured. Model results and lake-based hydroclimate reconstructions display very different temporal patterns over the last millennium. Additionally, there is no common signal among the model time series, at least until 1850. This suggests that simulated hydroclimate fluctuations are mostly driven by internal variability rather than by common external forcing. After that, half of the models used simulate a relatively clear response to forcing, but this response is different

between the models. Overall, the link between precipitation and tropical sea surface temperatures (SSTs) over the pre-industrial portion of the last millennium is stronger and more robust for the Challa/Naivasha region than for the Masoko/Malawi region. At the inter-annual time scale, last-millennium Challa/Naivasha precipitation is positively (negatively) correlated with western (eastern) Indian Ocean SST, while the influence of the Pacific Ocean appears weak and unclear. Although most often not significant, the same pattern of correlations between the East African rainfall and the Indian Ocean SST is still visible when

using the last-millennium time series smoothed to highlight centennial variability, but only in fixed-forcing simulations. This means that, at the centennial time scale, the effect of (natural) climate forcing can overwhelm internal climate variability in large-scale tele-connections.

## 1 Introduction

In 2011, the Horn of Africa was affected by the most serious drought in decades leading to severe humanitarian conse-

quences including food and water shortages, acute malnutrition, mass displacement and conflicts (OCHA, 2011; Hillbruner and Moloney, 2012). This drought was followed the next two years by strong pluvial events that triggered floods in Kenya and some parts of Somalia (OCHA, 2012; IFRC, 2013a, b). These consecutive and opposite extreme events illustrate the strong inter-annual variability characterizing East African rainfall (e.g. Nicholson et al., 2012). Due to the seasonal migration of the



ITCZ back and forth across the equator, East African rainfall has often a bimodal annual cycle with a main rainy season during March-May (often referred as long rains) and a weaker rainy season during October-December (often referred as short rains) (e.g. Yang et al., 2015). The short rains are more variable from one year to another, and thus drive most of the observed inter-annual variability (e.g. Hastenrath et al., 1993; Nicholson, 1996, 2014)). Numerous studies have emphasised the

5 tele-connection at this time scale between the East African short rains, the El Niño/Southern Oscillation (ENSO, e.g. Ogallo et al., 1988; Hastenrath et al., 1993; Nicholson and Selato, 2000; Schreck and Semazzi, 2004; Hoell et al., 2014) and the Indian Ocean Dipole (IOD, e.g. Goddard and Graham, 1999; Saji et al., 1999; Webster et al., 1999; Clark et al., 2003; Ummenhofer et al., 2009; Izumo et al., 2014), predominantly through atmospheric adjustments to the Walker circulation (e.g. Klein et al., 1999; Izumo et al., 2014). The 2011 drought, in particular, was triggered by a strong La Niña event (IRI, 2010; Hoell et al.,

2014) associated with a negative IOD (JAMSTEC, 2015) that resulted in drier than normal conditions in East Africa during the short rains at the end of 2010. The drought then worsened due to subsequent failure of the long rains in 2011 (March to May, Lyon and DeWitt, 2012).

Unlike short rains, the tele-connection between inter-annual East African long rains and the Indian and Pacific oceans is generally considered to be weak (e.g. Mutai and Ward, 2000; Pohl and Camberlin, 2006; Nicholson, 2014). However, failure

of the long rains in 2011, which is part of a substantial decline in long rains that started around 1999 (Lyon and DeWitt, 2012) or even earlier (Funk et al., 2008), was associated with abrupt warming of the Indian Ocean (Funk et al., 2008; Williams and Funk, 2011) or of the western tropical Pacific (Merrifield, 2011; Lyon and DeWitt, 2012), also altering the Walker circulation. Merrifield (2011) and Lyon and DeWitt (2012) do not propose a specific cause for this warming. By contrast, Funk et al. (2008) and Williams and Funk (2011) attribute the large-scale shift in Indian Ocean SSTs to anthropogenic forcing, and suggest that

a warmer future climate will bring an increased frequency of drought conditions in tropical eastern Africa owing to further reductions in the long rains.

This hypothesis is at odds with general circulation models (GCMs) showing that a warmer climate is associated with an enhanced pattern of precipitation minus evaporation (P-E) in the intertropical convergence zone (ITCZ), including the East African long rains (Seager et al., 2010; Laîné et al., 2014). Other model-based studies (McHugh, 2005; Shongwe et al., 2011;

Kirtman et al., 2013) furthermore support that increasingly wetter rainy seasons in East Africa are due to a weakening of the Walker circulation, likely linked to anthropogenic warming (Vecchi et al., 2006; Vecchi and Soden, 2007). This is corroborated by some observations which appear to show a weakening of the Walker circulation already since the mid-19th century (Vecchi et al., 2006). However, there is no agreement on that in observation-based studies, as illustrated by L'Heureux et al. (2013) who found a multi-decadal strengthening of the Walker circulation in the same period. In addition, some regional climate models

forced at the boundaries by ensemble-mean GCM results are not consistent with GCMs in that they show a reduction rather than increase in the long rains (Vizy and Cook, 2012; Cook and Vizy, 2013).

The poor ability of the GCMs in simulating the observed recent downward trend of the long rains implies either that it is part of (multi-)decadal natural variability (Lyon and DeWitt, 2012; Yang et al., 2014; Lyon, 2014), or that the GCMs inadequately represent the climate processes occurring in the region. In this context it is crucial to assess the performance of GCMs in

simulating East African rainfall. Yet studies in which this has been done (e.g. Conway et al., 2007; Anyah and Qiu, 2012;



Otieno and Anyah, 2012; Yang et al., 2014) reached contrasting conclusions depending on the region or spatial scale, or on the variables and models considered. Specifically, they showed that the mean seasonal cycle of precipitation is reasonably well simulated by the majority of GCMs, but that there is a large spread among them in capturing the actual dominant peaks where rainfall is bimodal (Anyah and Qiu, 2012; Otieno and Anyah, 2012). Additionally, most models appear to have significant

biases in monthly mean precipitation, and the observed link between East African rainfall and Indian Ocean SST is often not well represented (Conway et al., 2007; Yang et al., 2014).

All the above studies are limited to the recent past where direct measurements of precipitation exist. However, the period considered, which ranges from the last few decades to 150 years at most, is not sufficient to capture the multi-decadal variability that is thought to be an important component of East African hydroclimate (Verschuren, 2004; Tierney et al., 2013). Therefore,

to complement those studies our goal here is to extend this analysis to the last millennium by analyzing proxy records of past hydroclimatic change over this period in conjunction with simulations performed in the framework of the third phase of the Paleoclimate Modelling Intercomparison Project (PMIP3; Otto-Bliesner et al., 2009) and of the fifth phase of Coupled Model Intercomparison Project (CMIP5; Taylor et al., 2012). As for available hydroclimate proxy records, all reconstructions that are both well-dated and span the last millennium with sufficient time resolution to be compared with model results are based

on lake-sediment records (Verschuren, 2004). In this study, we consider proxy records describing the water-balance history of Lake Challa and Lake Naivasha in eastern equatorial Africa, and of Lake Masoko and Lake Malawi in southeastern (but still inter-tropical) Africa (Fig. 1). These four lake records are part of the East African hydroclimate synthesis recently achieved by Tierney et al. (2013), and all characterized by strong multi-decadal to centennial variability. As a consequence, the present study focuses on relatively long-term hydroclimate changes, and on variation in annual means rather than individual rainy

seasons, in contrast with model and observation-based studies about variation in recent East African rainfall (e.g. Anyah and Qiu, 2012; Yang et al., 2014).

Specifically, our study aimed to investigate GCM performance in simulating reconstructed long-term hydroclimate change throughout the last millennium; to define whether the simulated changes reflect the forcing of external climate drivers; and to assess the stability of large-scale tele-connections between East African rainfall and global SSTs. This paper is structured

as follows. In Section 2 we introduce the model experiments and proxy-based reconstructions. In Section 3 we evaluate the comparative performance of six different GCMs to simulate the seasonal cycle of rainfall over the study regions and its tele-connection to tropical SSTs, over the recent period. In Section 4 we analyze the results of model simulations spanning the last millennium. The contribution of forced and internal variability on those simulated hydroclimate changes and the stability of large-scale tele-connections are finally investigated in Section 5, followed by a discussion and conclusions in Section 6.

## 2   Data and methods

### 2.1   PMIP3/CMIP5 model experiments

Climate model simulations from PMIP3 (Otto-Bliesner et al., 2009) and CMIP5 experiments (Taylor et al., 2012) were obtained from the Program for Climate Model Diagnosis and Inter-comparison (PCMDI; http://pcmdi9.llnl.gov) and the Earth System



Grid (www.earthsystemgrid.org) archives. The six GCMs selected (Table 1) are those for which the diagnostic variables of interest, i.e. precipitation, actual evaporation and SST, were available at the time of our analysis, for both *past1000* (850–1850 AD) and *historical* (1850–2005) periods, as well as for *pre-industrial control* runs.

**Table 1.** Modeling centers, parameters and references of the CMIP5/PMIP3 models used in this study.

| Model name | Institution | Resolution (lat*lon) | | Ensemble members | | Reference |
| --- | --- | --- | --- | --- | --- | --- |
| | | Ocean | Atm. | past1000 | historical | |
| CCSM4 | National Center for Atmospheric Research | 384*320 | 192*288 | 1 | 6 | Gent et al. (2011) |
| CESM1 | National Center for Atmospheric Research | 384*320 | 96*144 | 10 | 10 | Otto-Bliesner et al. (2015) |
| GISS-E2-R | NASA Goddard Institute for Space Studies | 90*144 | 90*144 | 1 | 6 | Schmidt et al. (2014) |
| IPSL-CM5A-LR | Institut Pierre-Simon Laplace | 149*182 | 96*96 | 1 | 5 | Dufresne et al. (2013) |
| MPI-ESM-P | Max Planck Institute for Meteorology | 220*256 | 96*192 | 1 | 2 | Stevens et al. (2013) |
| BCC-CSM1-1 | Beijing Climate Center, China Meteorological Administration | 232*360 | 64*128 | 1 | 3 | Wu et al. (2014) |

Although not continuous, except for CESM1 and MPI-ESM-P, the first ensemble members (r1i1p1) of *past1000* runs were

merged with the corresponding *historical* simulations to obtain results spanning the period 850–2005 AD. The impact of the discontinuity is probably limited since it falls within the range of internal climate variability for surface variables in all cases. The simulations are driven through the last millennium by both natural (orbital, solar, volcanic) and anthropogenic (well-mixed greenhouse gases, ozone, tropospheric aerosols, land use) sources of climate forcing. Earth's orbital parameters vary according to the calculations of Berger (1978), except in CESM1 where they are held constant at 1990 values. Depending on

the model, different reconstructions of solar irradiance variability are applied. All models except CESM1 and GISS-E2-R use the reconstruction by Vieira and Solanki (2009) from 850 to 1609 and by Wang et al. (2005) from 1610 to 2005. CESM1 uses the reconstruction by Vieira et al. (2011) with the spectral variations and '11-year' solar cycle from Schmidt et al. (2012), whereas GISS-E2-R uses the Steinhilber et al. (2009) reconstruction until 1849 and the Wang et al. (2005) reconstruction from 1850 onwards.

The forcing related to volcanic aerosols is derived from Crowley and Unterman (2013) in GISS-E2-R and MPI-ESM-P and from Gao et al. (2008) in the four other GCMs. Reconstructed and observed changes in the major greenhouse gases driving *past1000* (Flückiger et al., 2002; MacFarling Meure et al., 2006) and *historical* (Hansen and Sato, 2004) simulations, ie. $CO_2$, $CH_4$ and $N_2O$, are the same in all models. Anthropogenic changes in land use/land cover over the last millennium are based on the reconstruction of Pongratz et al. (2008) in MPI-ESM-P, of Pongratz et al. (2008) followed by the reconstruction of

Klein Goldewijk and van Drecht (2006) for the historical period in GISS-E2-R and of Pongratz et al. (2009) followed by the



reconstruction of Hurtt et al. (2011) for historical time in CCSM4 and CESM1. BCC-CSM1-1 and IPSL-CM5A-LR do not consider any change in land use/land cover, whose distribution is fixed to the pre-industrial value. Tropospheric ozone and aerosols variations are also taken into account in *historical* experiments in CCSM4, CESM1, GISS-E2-R, MPI-ESM-P and BCC-CSM1-1, and are all based on the dataset described in Lamarque et al. (2010). These changes are however neglected in
IPSL-CM5A-LR, which considers a constant concentration fixed at the pre-industrial level. More information on the climate-forcing reconstructions used for driving *past1000* experiments and on their implementation can be found in Schmidt et al. (2011) and Schmidt et al. (2012).

## 2.2   Proxy-based hydroclimate reconstructions

Despite the relatively large number of lakes present across East Africa, only a small sub-set of them combine continuous
sedimentation (and thus, archiving) with hydrological sensitivity to climatic moisture-balance changes (Verschuren, 2003). Recently, Tierney et al. (2013) selected seven East African proxy records that are well-dated and meet the criteria of primarily reflecting hydroclimate variation and of covering the last millennium with a time resolution better than 50 years. Our present study focuses on four of these records, originating from Lake Challa, Lake Naivasha, Lake Masoko and Lake Malawi. The records from Lake Tanganyika and Lake Edward are not considered because they are located far from the Indian Ocean and
thus to the west of our region of interest. The record from Lake Victoria is not considered because the representation of this large lake varies from one model to another, which precludes a meaningful comparison between model results and proxy records for this site. Indeed, while most GCMs ignore its presence, the MPI-ESM-P model specifically recognizes it as a surface-water body surrounded by continent. It is well known that Lake Victoria itself strongly influences the regional hydrological cycle (e.g. Thiery et al., 2015), but those effects cannot be adequately reproduced in GCMs with relatively coarse spatial resolution.
The 1000-year time series representing hydroclimate variation in the Lake Challa region in Tierney et al. (2013) is the first principal component of composite variation in three moisture-balance proxies, namely a presumed indicator of catchment runoff (the branched and isoprenoidal tetraether index (BIT); Verschuren et al., 2009), an isotopic proxy for rainfall source and intensity ($\delta$D in the leaf waxes of terrestrial plants; Tierney et al., 2011), and a proxy for variation in dry-season length and windiness (varve thickness; Wolff et al., 2011). The time series from Lake Naivasha is a lake-level reconstruction based
on sediment lithostratigraphy (Verschuren, 2001), supported by salinity reconstructions based on fossil diatom and midge assemblages (Verschuren et al., 2000). The hydroclimate record from Lake Masoko is inferred from the low-field magnetic susceptibility of the sediment, which is a proxy for lake-level changes and/or wind stress. Two such records are available for this lake, one that goes back to -43,300 AD (Garcin et al., 2006) and one that starts around 1500 AD (Garcin et al., 2007). The Masoko time series in this paper is obtained from Tierney et al. (2013), who used the last millennium of the longer record but
with age-depth tie-points translated from the shorter record (Anchukaitis and Tierney, 2012). Finally, the hydroclimate record from Lake Malawi is inferred from the mass accumulation rate of the terrigenous sediment fraction, presumed to be a runoff proxy (Brown and Johnson, 2005; Johnson and McCave, 2008). Table S1 provides more information about each of these four proxy records.





## 3 Evaluation of model performance over the period 1979-2005

### 3.1 Mean seasonal cycle of precipitation

This section assesses the ability of various GCMs to reproduce the observed mean state and seasonal cycle of East African rainfall. Although similar analyses have been performed previously (e.g. Conway et al., 2007; Anyah and Qiu, 2012; Otieno and Anyah, 2012; Yang et al., 2014), it is important to repeat it here focusing specifically on the areas where our four study sites are located (Fig. 1).

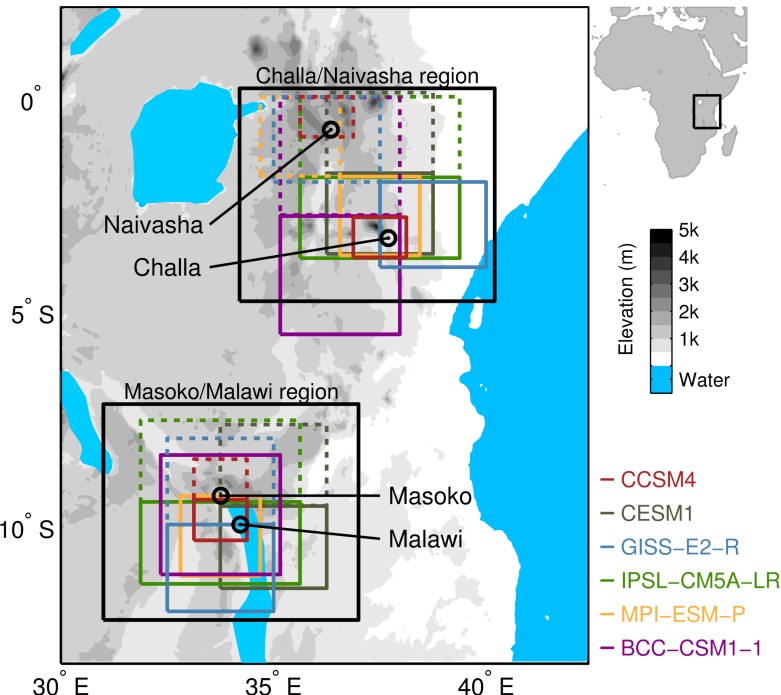

**Figure 1.** Location of lakes Challa, Naivasha, Masoko and Malawi in East Africa, along with the individual grid cells containing each of these four sites in six different CMIP5 models (dashed (plain) boxes for the grid cells that contain the lakes Naivasha and Masoko (Challa and Malawi)), and the two larger regional domains over equatorial East Africa (Challa/Naivasha region) and southeastern inter-tropical Africa (Masoko/Malawi region). Topography from Amante and Eakins (2009).

The number of rain-gauge stations in East Africa is small, and the observations suffer both from an uneven spatial distribution and from gaps in time due to maintenance issues (Dinku et al., 2007; Sylla et al., 2013). We have therefore used global gridded datasets that merge the information from rain-gauge stations, remote sensing, and/or reanalysis results. However, these gridded datasets have their own uncertainties, related for instance to the number and the treatment of rain-gauge measurements or of radar precipitation estimates (Otieno and Anyah, 2013). In order to estimate the effect of these uncertainties on our conclusions, we used four global monthly gridded datasets of precipitation. Version 6 of the Global Precipitation Climatology Centre data set (GPCC-v6; Schneider et al., 2014) is a reanalysis using rain-gauge data only. It spans the period from 1901 to the present



with a spatial resolution of 0.5°×0.5°. Version 2.2 of the Global Precipitation Climatology Project (GPCP-v2.2; Huffman et al., 2009) combines rain-gauge and satellite-based precipitation data on a 2.5°×2.5° grid from 1979 to the present. The Climate Prediction Center (CPC) Merged Analysis of Precipitation data set (CMAP; Xie and Arkin, 1997) covers the same period at identical spatial resolution but combines rain-gauge data and different satellite estimates with NCEP/NCAR reanalysis results

in gaps. Finally, NOAA's Precipitation Reconstruction over Land data set (PREC/L Chen et al., 2002) is based only on rain-gauge data and covers the period from 1948 to the present with a spatial resolution of 0.5°×0.5°.

Due to the seasonal north-south migration of the Intertropical Convergence Zone (ITCZ) across the equator, East African rainfall is characterized by a strong annual cycle that differs from one location to another (Nicholson, 1996; Otieno and Anyah, 2013; Yang et al., 2015). The seasonality of rainfall is bimodal over equatorial sites such as Lake Challa and Lake Naivasha,

with two rainy seasons occurring in March to May (long rains) and in October-December (short rains), while it is unimodal in the southeastern lakes Masoko and Malawi, with a maximum between November and April during the austral summer (bar charts in Fig. 2). The observed mean monthly rainfall in each of the four study areas over the period 1979-2005 serves as our reference frame to compare the success of individual GCMs in simulating the present-day seasonal cycle, using model results from the individual grid cells which include the lakes (Fig. 1).

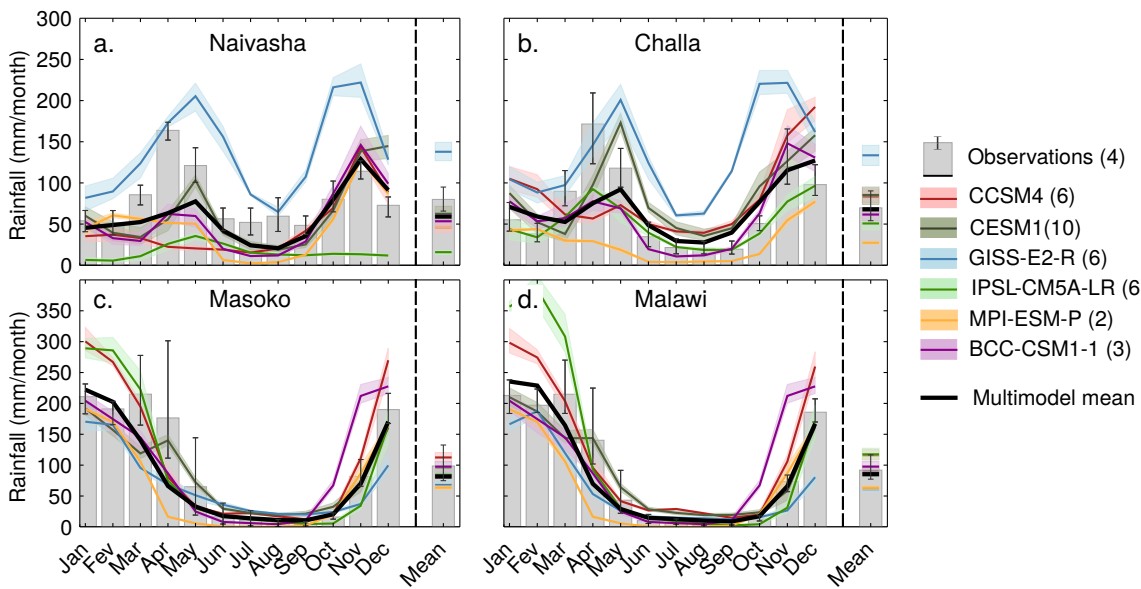

**Figure 2.** Mean monthly rainfall over lakes Challa, Naivasha, Masoko and Malawi (a-d) in observations (bar plots) and in six CMIP5 models (curves) over the period 1979-2005. The number of datasets (for the observations) or ensemble members (for the models) used in each case is shown in brackets. Error bars and shaded areas represent the range of those observations or ensemble member values.

For Lake Naivasha (Fig. 2a), simulated monthly mean precipitation is characterized by a large spread among models, CESM1 being the only one approaching the observations. Observed short rains over this lake are relatively well simulated by most models, except by GISS-E2-R and IPSL-CM5A-LR which respectively strongly overestimate and underestimate them. During



the other seasons, simulated precipitation over Lake Naivashais underestimated by all models except GISS-E2-R, including during the main rainy season in boreal spring. For Lake Challa (Fig. 2b), three models are relatively consistent with the observed values of monthly mean precipitation: CCSM4, CESM1 and BCC-CSM1-1. Most models underestimate the long rains except CESM1 and GISS-E2-R, although in these two models this rainy season is delayed by one month compared to observations.

In contrast with the observations all models show highest rainfall in October or November rather than April, but the spread is again large. These differences between models during the short rains are consistent with biases at larger spatial scales noted in previous studies of the East African region. Anyah and Qiu (2012) and Yang et al. (2014) indeed showed that, despite a large spread among CMIP3 and CMIP5 models, most of them tend to underestimate and to shift by one month the long rains, and to overestimate the short rains. Agreement among models and between models and observations is much higher in Lake Masoko

and Malawi (Fig. 2c-d). Both the rainy and dry seasons are well simulated, although the amount of rain is overestimated by CCSM4 and IPSL-CM5A-LR during the rainy season. Overall, climate models used in this study are thus able to represent the unimodal rainfall seasonality characterizing the region encompassing lakes Masoko and Malawi, while the magnitude of the two rainy seasons characterizing the region encompassing lakes Challa and Naivasha is less well captured.

Comparing model results and data at individual grid cells can be questionable since model skill at this scale is often very

limited. For instance, a small shift in the spatial structure of the simulations compared to the observations can lead to large difference in precipitation. Local topographic features not well represented at the grid scale may also have a significant impact. Besides, the surface areas of the grid cells which include the lake sites strongly differ from one model to another (Fig. 1). To get rid of the problems linked to differences in spatial resolution and to remove local noise in favour of more regional patterns, it seems better to take into account, instead of individual grid cells, larger regions presenting common characteristics as discussed

in the next section.

### 3.2 Link between individual grid cells and the larger spatial domains selected for analysis

Since Lake Challa and Lake Naivasha on the one hand and Lake Masoko and Lake Malawi on the other have a very similar climatology and seasonal cycle both in models and observations (Fig. 2), we consider a spatial domain which includes the first two lakes 0.2° N to 4.8° S and 34.2° E to 40.2°E , referred to as the Challa/Naivasha region) and a second one which includes

the last two ( 7.2° S to 12.2°S and 31° E to 37°E, referred to as the Masoko/Malawi region; Fig. 1). However, using the larger grid boxes raises the issue of whether the proxy-based reconstructions employed to assess model performance through the last millennium are representative for these larger spatial domains.

Fig. 3 shows the correlation between modeled mean annual rainfall in the individual grid cells containing the four lake sites and the two larger domains defined for our analysis, for both the recent period and the last millennium. Changes in

precipitation within the grid cell containing Lake Masoko, within the grid cell containing Lake Malawi and within the larger box representing the Masoko/Malawi region are highly and significantly correlated for all models and observations, and for both periods. Rainfall over Lake Challa, Lake Naivasha and the Challa/Naivasha region is also positively correlated in both periods for each source of information, but note that the GISS-E2-R model depicts a relatively low but significant correlation between Lake Challa and Lake Naivasha over the last millennium.



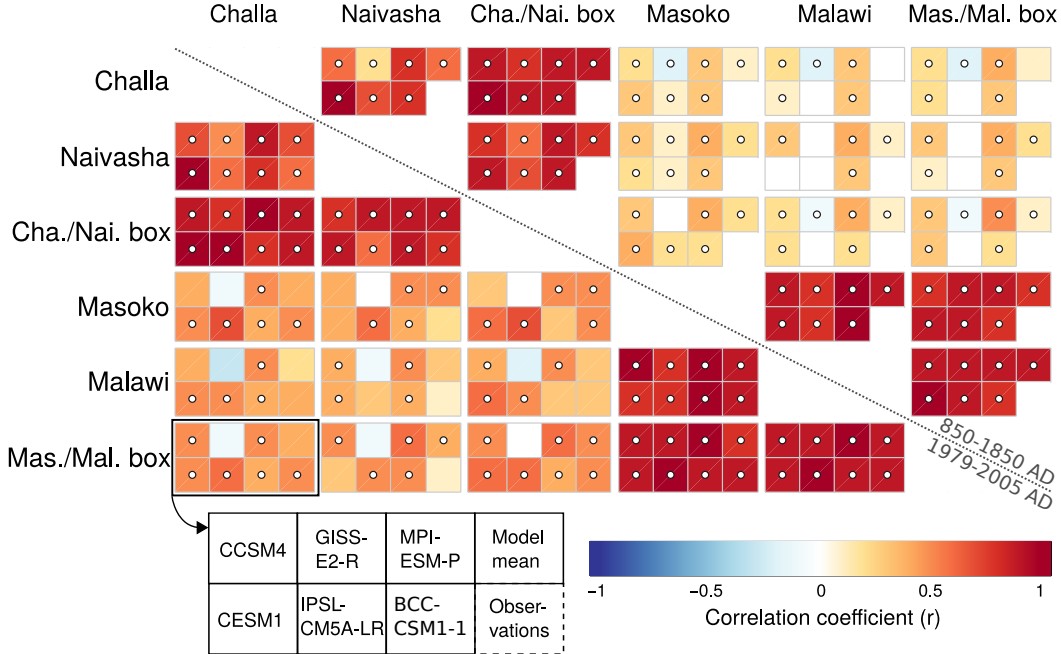

**Figure 3.** Pearson correlation coefficients between mean annual rainfall values at six different locations: the single GCM grid cells that contain lakes Challa, Naivasha, Masoko and Malawi, and the two larger grid boxes which delineate the Challa/Naivasha and Masoko/Malawi regions. The lower left half of the figure shows these correlations for the period 1979-2005 in the six GCMs considered in this study (CCSM4, CESM1, GISS-E2-R, IPSL-CM5A-LR, MPI-ESM-P, BCC-CSM1-1; only the first ensemble member r1i1p1 is considered) plus the model mean, and in the average of four gridded observation datasets (GPCC-v6, GPCP-v2.2, CMAP and PREC/L, see references in Section 3.1), in the order shown below the main panel. The upper right half of the figure shows the same results for the period 850-1850 AD (*past 1000*). Squares with a central white circle represent combinations for which the null hypothesis of no correlation can be rejected at the 5% level.

Observed recent rainfall over Lake Challa, Lake Naivasha, and the Challa/Naivasha region shows no or only a weakly positive relationship with rainfall over Lake Masoko, Lake Malawi and the Masoko/Malawi region, underscoring the climatic dichotomy characterizing East Africa. Note that no negative correlation is found except very weak ones in GISS-E2-R between rainfall over Lake Challa and rainfall over Lake Masoko, Lake Malawi and the Masoko/Malawi region. The dipole between
5   the eastern coastal 'Horn of Africa' region and the interior rift-valley region highlighted in Tierney et al. (2013) is thus not observed at the annual time scale considered here. Although the correlations are quite similar for the recent period and the last millennium, they are somewhat higher in the former, which may be because at the present-day anthropogenic forcing acts in the same way at every location within each model.

Overall, both observations and model results show that the two selected regions are on the one hand characterized by different
10   patterns of temporal variability and, on the other hand, representative of the individual grid cells containing the lake sites, not only regarding annual precipitation changes over the recent period and the last millennium, but also regarding annual mean absolute values and seasonal cycles (not shown).



### 3.3 Large-scale tele-connections

In addition to evaluation of the mean state, it is important to assess the ability of climate models to represent the observed climate dynamics associated with the inter-annual variability of East African rainfall. This is illustrated here by analysing the links between East African precipitation and tropical SSTs, an important characteristic of the system as mentioned in the

introduction. Two different SST datasets are used: version 3 of the Hadley Centre dataset (HadSST3; Kennedy et al., 2011a, b), based on in-situ measurements covering the period from 1850 to 2015 on a $5° \times 5°$ grid, and version 3b of the Extended Reconstructed Sea Surface Temperature dataset (ERSST-v3b; Smith et al., 2008), also based only on in-situ data, which covers the period 1854 to 2015 on a $2° \times 2°$ grid. Since the proxy-based reconstructions used in this study are considered to represent mean annual conditions, interest is here mainly on mean annual results. However, mean annual results for East Africa's hy-

droclimate represent a combined picture, as the relative amplitudes and the strength and character of tele-connections between East African rainfall and tropical SSTs at the inter-annual time scale are different from one season to another.

For both regions, the largest correlations between observed rainfall and SSTs are found during the boreal autumn (OND), with the well-known pattern of positive (negative) correlation between East African rainfall and western (eastern) Indian Ocean SSTs, as well as SSTs in the central/eastern (western) equatorial Pacific (Fig. S1 and S2 OND), which resembles the SST

pattern during an El Niño phase of ENSO and a positive phase of the IOD. This period corresponds to the short rain season in Challa/Naivasha and to the start of the single rainy season in Masoko/Malawi farther south (Fig. 2). For the other seasons, the spatial pattern of correlation differs between the two study areas, with the greatest difference occurring during boreal winter (JFM). In Challa/Naivasha, where rainfall is relatively low during this transition period between the short rains and the long rains, it appears to be positively correlated with SSTs over the central/north Indian Ocean (Fig. S1 JFM). In contrast, just a few

isolated significant correlations are found between rainfall in boreal winter over Masoko/Malawi and tropical SSTs (Fig. S2 JFM), although this period marks the annual maximum in precipitation (Fig.2). Tele-connections to tropical SSTs are weak in both regions during the boreal spring (AMJ; Fig. S1 and S2 AMJ), which corresponds to the long rains in Challa/Naivasha and to the end of the rainy season in Masoko/Malawi; and are also weak also during the boreal summer (JAS; Fig.S1 and S2 JAS), which is the principal dry season in both regions (Fig. 2).

When considering mean annual results, the correlation between observed precipitation over the Challa/Naivasha region and SSTs shows a pattern similar to that during the short rains, although damped, with positive correlation in the western Indian Ocean and central/eastern tropical Pacific and negative correlation centered on Indonesia (Fig. 4a left). In contrast, the correlations between Masoko/Malawi rainfall and SSTs after computing annual averages are mostly weak and non-significant in the Indian Ocean and negative over the Indonesian region while some zones in the Pacific show positive correlations (Fig.

4b left), even though the correlation pattern for boreal autumn (OND) rainfall is similar to that found using precipitation over Challa/Naivasha. In both cases, the spatial patterns of correlation derived from the different combinations of datasets show very similar results, with spatial correlations of the obtained patterns always exceeding 0.80.

Out of the six GCMs, only CESM1 and MPI-ESM-P correctly simulate the spatial pattern of observed Challa/Naivasha rainfall (Fig. 4a). Nevertheless, with values equal to 0.27 and 0.16 the significant correlation coefficients between, respectively,

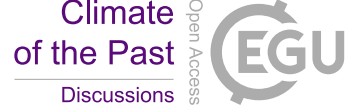

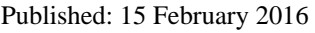

**Figure 4.** Pearson correlation coefficients between mean annual rainfall over the Challa/Naivasha region (a, upper three rows) or Masoko/Malawi region (b, lower three rows) and global SSTs in observations and climate models (here, only the first member r1i1p1 is used) for the period 1950-2000. In areas overprinted with white circles, a null hypothesis of no correlation can be rejected at the 5% level. No data are available from grey areas.

observed and simulated rainfall-SST correlation maps remain relatively low for these two models (Table S2). GISS-E2-R, IPSL-CM5A-LR and BCC-CSM1-1 display totally different patterns compared to observations, and CCSM4 even simulates



an opposite pattern, with positive (negative) correlations between simulated precipitation over the Challa/Naivashaarea and the eastern Indian Ocean (central/eastern Pacific).

Model results show a wide range of tele-connections between Masoko/Malawi rainfall and SSTs (Fig. 4b.). None of them shows the observed pattern of correlations, which is weak but seems robust given that it is similar regardless of the combina-

tions of datasets used. GISS-E2-R, IPSL-CM5A-LR and MPI-ESM-P patterns are positively, although poorly, correlated with the data (Table S2). However, the correlation coefficients obtained by GISS-E2-R and IPSL-CM5-LR are the result of a compensative effect between wrong seasonal tele-connection patterns (Fig. S2). CCSM4 simulates a strong relationship between Masoko/Malawi and Indian and Pacific Ocean SSTs, but in an opposite sign than expected, with negative correlations in the western Indian Ocean and central/eastern equatorial Pacific, and positive correlations in the eastern Indian Ocean. Although

CESM1 can simulate the pattern observed in boreal winter, no correlation is found when considering annual mean results.

Overall, most climate models fail to simulate observed tele-connections of Challa/Naivasha and of Masoko/Malawi rainfall to large-scale SST patterns at the inter-annual scale. Annual smoothing makes these tele-connections complex and of relatively limited magnitude, especially for Masoko/Malawi rainfall. However, this cannot by itself explain the low model performance since their skill is not greatly improved when only considering the OND rainfall, shown to be strongly and robustly corre-

lated to Indian and Pacific SSTs in observations. Only CESM1, MPI-ESM-P and, to a lesser extent, BCC-CSM1-1 simulate the observed patterns for both regions during this season (Table S2). Inconsistent with the observations, IPSL-CM5A-LR and GISS-E2-R simulate a relatively homogeneous tele-connection pattern throughout the year, while the strong seasonality depicted by CCSM4 has almost everywhere incorrect signs. MPI-ESM-P and CESM1 thus tend to stand out by their positive correlation with annual observations in both regions and only in Challa/Naivasha, respectively, for theright reasons. These mit-

igated results are, to some extent, consistent with the study of Rowell (2013). Indeed, although using different methodologies and selected areas, this author has shown that the tele-connections between rainfall over central East Africa (roughly corresponding to the Masoko/Malawi region) and both equatorial Pacific and Indian Ocean SSTs are particularly hard for CMIP5 models to simulate. Furthermore, Rowell (2013) reached a similar conclusion about the tele-connections between rainfall over the greater Horn of Africa region, which includes the Challa/Naivasha region, and equatorial Pacific SSTs.

## 25    4    Reconstructed and simulated hydroclimate over the last millennium

### 4.1    Hydroclimate changes deduced from proxy-based reconstructions

Making abstraction of chronological uncertainties, the Challa and Naivasha proxy records show discrepancy during the first four centuries of the last millennium. Indeed, the former roughly shows a drying trend while the opposite is recorded in the latter (Fig. 5). From around 1400 AD, however, the main hydroclimate trends inferred from those two records are similar: both

show relatively dry conditions followed by a wetting trend peaking between about 1700 and 1750. After this peak in humidity, both hydroclimate reconstructions depict an abrupt transition towards a relatively short drier period in the early 19th century, followed by smaller-scale hydroclimate fluctuations in Lake Naivasha and a clear wetting trend in Lake Challa.




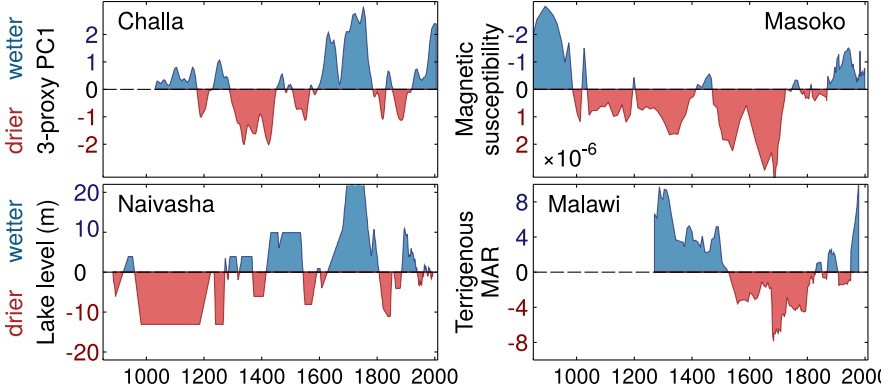

**Figure 5.** Lake-based moisture-balance reconstructions used in this study; references with details on the proxies are provided in Section 2.2. Ordinate axes for each proxy are oriented such that wetter (drier) hydroclimate conditions point upward (downward).

Lake Masoko and Lake Malawi are also characterized by similar long-term hydroclimate changes from about 1400, but which contrast with the Challa/Naivasha pattern. Indeed, the reconstructions agree on a drying trend culminating around 1700, before an increase in humidity towards the present.

### 4.2 Interpretation of proxy-based reconstructions from a model perspective

Lakes are complex hydrological systems. To understand their dynamics it is necessary to account for the inflow and outflow from rivers and surface runoff, rainfall on the lake surface, evaporation from the lake surface, groundwater inflow and/or outflow, as well as interactions with the aquifer surrounding the lake (e.g. Becht and Harper, 2002). Besides, these processes are not represented directly in relatively coarse-resolution GCMs. These models simulate the large-scale moisture balance, i.e. precipitation minus actual evaporation (P-E), the latter depending on potential evaporation and soil moisture content. Each of the sedimentary proxies used in the lake-based climate reconstructions can by qualitatively interpreted as smoothed versions of the local-to-regional climatic moisture balance, at least when considered on multi-decadal to centennial time scales. P-E is thus the model variable that has been chosen here for comparison with the reconstructed histories of lake-level fluctuation, catchment runoff or drought-season severity, depending on the lake (see Section 2.2). This section describes the relative contributions of rainfall and of evaporation in P-E, which allows knowing whether the respective regions containing each record are more influenced by precipitation or evaporation.

It was shown in Section 3.1 and in Section 3.2 that, despite their relative proximity, the two spatial domains used in this study, Challa/Naivasha and Masoko/Malawi, are quite different in terms of precipitation amount and seasonality as well as in precipitation trends through time.The study of P-E balance confirms that. Indeed, climate models simulate mean P-E values over Challa/Naivasha which are close to zero throughout the last millennium (Fig. 6a). In contrast, all models except GISS-E2-R show positive P-E values for the Masoko/Malawi region receives during the last millennium (Fig. 6b). The higher P-E for



Masoko/Malawi compared to Challa/Naivasha is mainly due to higher precipitation, which more than compensates for higher evaporation and thus leads to larger river runoff towards the lakes.

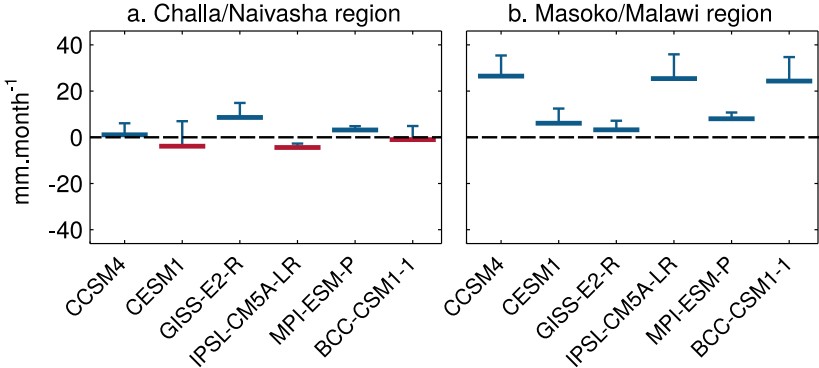

**Figure 6.** Mean annual precipitation minus mean annual evaporation (horizontal bars), and standard deviation of annual precipitation minus the standard deviation of annual evaporation (vertical bars) using the entire last-millennium results (850–2005) for each GCM. Red (blue) color means that evaporation (precipitation) dominates.

If we consider the standard deviation of variation in P and E through the last millennium, there is a consensus among models that P is more variable from year to year than E, for both regions (vertical bars in Fig. 6, which all plot upwards). This implies that most changes of P-E over time are due to changes in precipitation. Differences between the standard deviation of P and of E are also generally greater in Masoko/Malawi. This can be explained in all models except GISS-E2-R by a weaker relationship between P and E in this region than in Challa/Naivasha, as evidenced by weaker correlation coefficients (Table 2). Higher correlations indeed mean that a change in P is more often accompanied by similar change in E, which tends to bring variances closer.

**Table 2.** Pearson correlation coefficients between simulated rainfall and simulated evaporation in the Challa/Naivasha and Masoko/Malawi regions over the pre-industrial portion of the last millennium (850-1850 AD).

|  | Challa/Naivasha | Masoko/Malawi |
|---|---|---|
| **CCSM4** | 0.66* | 0.09* |
| **CESM1** | 0.57* | 0.35* |
| **GISS-E2-R** | 0.71* | 0.95* |
| **IPSL-CM5A-LR** | 0.91* | 0.69* |
| **MPI-ESM-P** | 0.68* | 0.62* |
| **BCC-CSM1-1** | 0.88* | 0.70* |

* significant at the 5% level





These results are robust throughout the last millennium, since the ratios between variable averages and variable variances remain relatively stable over time, even during the recent past where one could expect evaporation to increase relative to evaporation due to anthropogenic warming (not shown).

### 4.3 Comparison between reconstructions and model results

As briefly discussed in Section 4.2, the link between simulated and reconstructed variables is only indirect, which means that the magnitude of simulated P-E that should best fit the reconstructions is unknown. In consequence, the focus in this comparison is on common relative changes, rather than on their magnitude in absolute values. For better readability of the figures, each time series has been linearly standardized so that the maximum of the absolute values equals 1. A 100-year smoothing is applied to the model results in order to resemble temporal variability in the reconstructions.

Despite this smoothing, most model curves do not show any distinct long-term trend in the past millennium similar to that of the proxy-based reconstructions, and have much weaker fluctuations at the centennial scale than the reconstructions (Fig. 7). The correlation coefficients between model results and reconstructions, computed annually using interpolations for the reconstructed time series, are presented in Table S3. Most coefficients are low and non-significant at the 0.05% level, and can be for a same site either positive or negative depending on the models, which implies that there is no common signal between
models and data but neither among models.





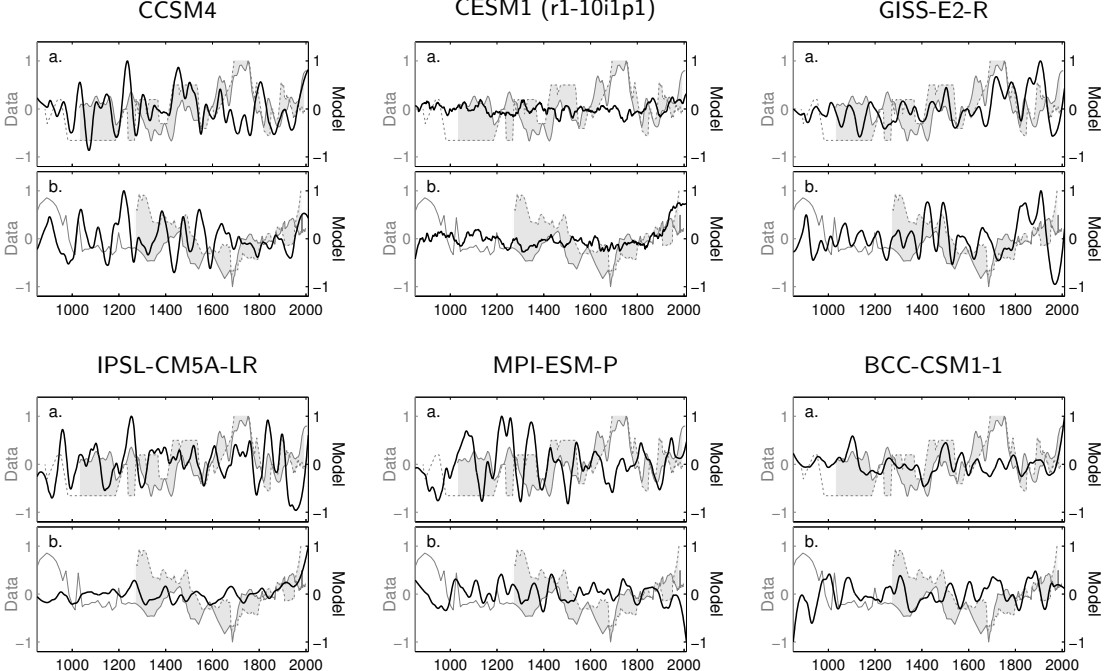

**Figure 7.** Comparison between last-millennium time series of the reconstructions (in grey) and of P-E simulated by six GCMs (in black) averaged over the Challa/Naivasha region (a), with the Naivasha record shown as dashed line and the Challa record as solid line; and over the Masoko/Malawi region (b), with the Malawi record shown as dashed line and the Masoko record as solid line. In both regions, the area between the between the two records is shaded in light grey. Both proxy-based and simulated time series are presented as anomalies with respect to the whole period, and are linearly standardized so that the absolute maximum equals 1. Ordinate axes are oriented such that wetter (drier) conditions point upwards (downwards). Model time series are annual mean values filtered using a loess method with a window of 100 years. For the CESM1 model, the black curve is the median of the ten ensemble members previously standardized and smoothed.

No individual site is characterized by an overall better agreement between model and data since the averages of correlation coefficients for each lake are close to zero. Furthermore, taking the four sites into account, no model appears to match substantially better the reconstructions than another. Nevertheless, some isolated positive correlations should be mentioned: CESM1 shows one positively and statistically significant correlation of 0.43 with the average of the reconstructions from Masoko and

5    Malawi, and GISS-E2-R correlates with the average Challa/Naivasha time series with a coefficient of 0.38.

This general model-data mismatch could arise from several reasons. First, as discussed in Section 3.2, the results from climate models are selected from the two regions Challa/Naivasha and Masoko/Malawi, that do not necessarily match the spatial representativity of proxy data, or may have trouble in representing the regional atmospheric dynamics responsible for changes in lake hydrology. Note that we consider only the individual grid cells which contain the proxy data sites, the correlation coef-

10    ficients are not substantially affected (not shown). This is consistent with the high correlations found for each model simulation between the individual grid cells and the larger spatial domain which contains them (Fig. 3). Second, the variables compared are different. Here, simulated regional P-E is indeed confronted with reconstructed lake level, catchment runoff, or seasonal



drought severity depending on the site (see Section 2.2). P-E, which mostly depends on variation in precipitation (see Section 4.2), is certainly related to these reconstructed variables, but sometimes in an indirect way that is difficult to assess precisely. Third, our model results only consider the immediate effect of P-E and do not account for long-term effects of a change in P-E. However, lake level during a particular year strongly depends on lake level during previous years. To address this issue,

we applied a first-order autoregressive model (AR-1) applied to each simulated time series. The AR-1 process is a simple persistence model where a realization of the system depends on the value at one time step earlier. This thus allows emulating a system with a chosen amount of memory. However, although this produces time series with low-frequency changes that are similar to the reconstructions, general model-data agreement is not substantially improved (Section A3).

The fact that all the model results appear different raises the question whether external forcing has any impact on the
10 simulated hydroclimate, forcing that is comparable between models (see Section 2.1) and is thus expected to put a comparable imprint on all time series. The lack of common timing among models in the simulated hydroclimate fluctuations for East Africa indeed suggests that they are mostly or exclusively driven by variability internal to the climate system.

## 5 The contribution of forced and internal variability in simulated hydroclimate changes

### 5.1 Hydroclimate changes over the past millennium

Whether the simulated East African hydroclimate results from internal variability or from changes in forcing is assessed in two steps: first by investigating potential common signals among models, and second by comparing variability of the last-millennium hydroclimate changes against control simulations. In the previous section we suggested that little or no link can be established between last-millennium hydroclimate changes simulated by the different models. Indeed, if we first consider the Challa/Naivasha region, most correlations between P-E time series simulated by different models are not statistically significant
and close to zero (Fig. 8). Actually, the fact that there is no positive correlation between hydroclimate curves from different models does not necessarily mean that there is no impact of forcing in each model. Indeed, the effect of changes in forcing may be different from one GCM to another, especially at the still relatively small spatial scale considered in this study. In this regard, it is of interest to note that for the one model for which multiple ensemble members were available (CESM1), there is also no correlation between the different ensemble members that differ only from slightly different air temperature at the
start of the experiments (Otto-Bliesner et al., 2015). This means that the forced response has a much smaller magnitude than internal variability for P-E in that region, evenat the multi-decadal time scale.

For the Masoko/Malawi region, the link between the hydroclimate time series produced by different models is also low. However, most ensemble members of CESM1 show significantly positive correlations with each other, mainly due to a common increase in P-E around 1800. This suggests an impact, although limited, of external forcing on Masoko/Malawi hydroclimate
during the last two centuries, as simulated by CESM1. For the other models, if a significant response to forcing is present, it is too different between themto be revealed by the correlation, except for IPSL-CM5A-LR which correlates positively with the ensemble members of CESM1.




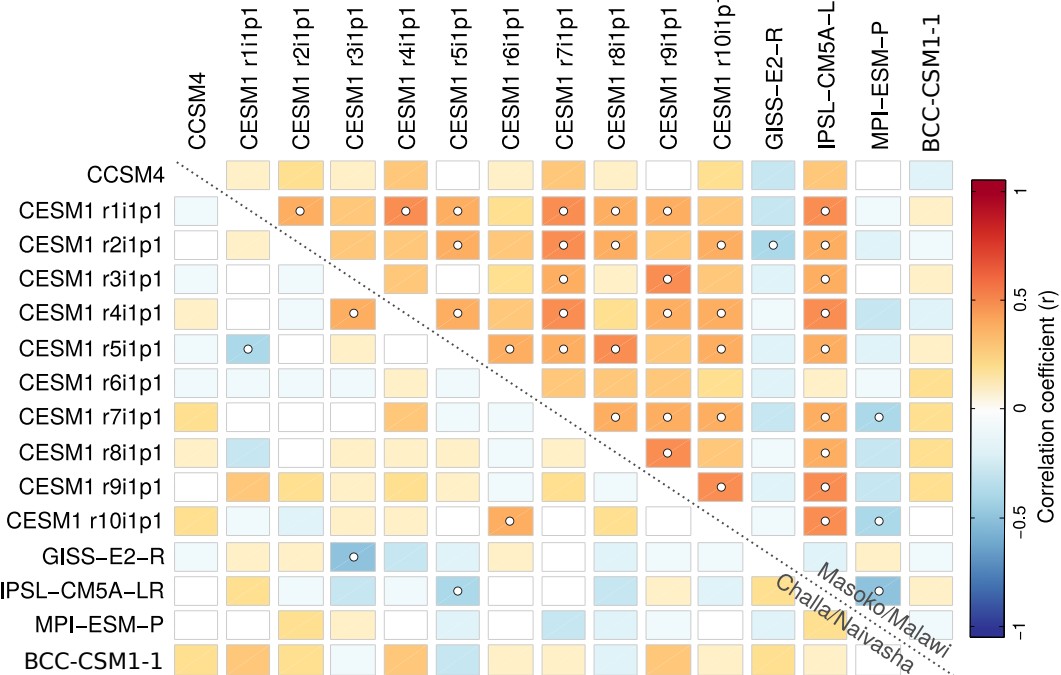

**Figure 8.** Pearson correlation coefficients betweenannual mean P-E over the last millennium (850-2005) as simulated by the different GCMs. The results were filtered using a loess method with a window of 100 years. The lower left half of the figure shows the results for the Challa/Naivasha region, the upper right half the results for Masoko/Malawi. Squares with a central white circle represent combinations for which the null hypothesis of no correlation can be rejected at the 5% level.

To complement this diagnostic, Fig. 9 shows P-E time series for the two regions from forced simulations, along with the variability in these time series and in the *pre-industrial control* runs, represented by the ±2 standard-deviation envelope. It shows that the P-E variance simulated over the last millennium is in most cases very similar to that of the control simulations. This indicates that the radiative changes from GHGs, solar variability and relatively short-lived volcanic effects have little

5    influence onthe selected variables compared to internal variability. The only noticeable difference in these variances is found for Masoko/Malawi in two of the models, CEMS1 and IPSL-CM5A-LR, which both show a P-E increase in recent time. However, in agreement with the results of previous studies (e.g. Hoerling et al., 2006) no significant common regional P-E patterns are found in these models (not shown). The similarity of the variances of P-E in fixed and time-varying forcing experiments, along with the lack of common P-E changes in the last-millennium simulations among models, obviates the

10    possibility of general agreement with the proxy results, and suggests that any apparent proxy-model agreement for this time period and these regions is coincidental. To obtain more information on the processes driving the simulated hydroclimate changes over the last millennium, the next section deals with the stability of tele-connections at different frequencies, and with the impact of changes in forcing on those tele-connections.





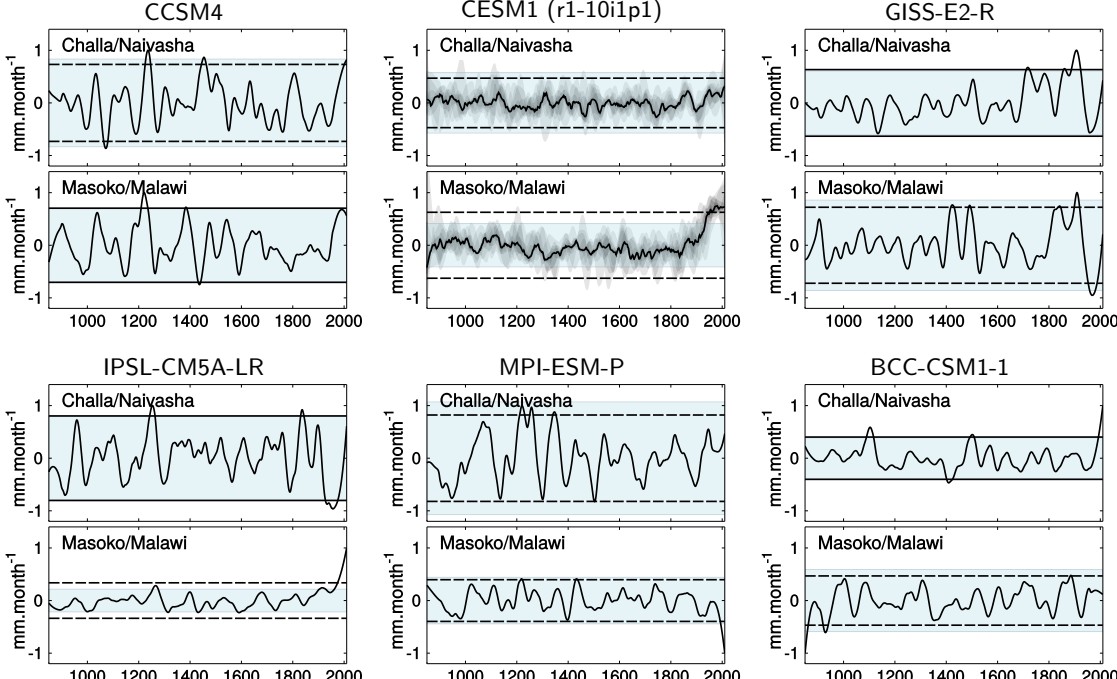

**Figure 9.** Simulated time series of P-E over the Challa/Naivasha and Masoko/Malawi regions throughout the last millennium (850–2005). Results are mean annual values smoothed using a loess filter with a window of 100 years and are presented as anomalies with respect to the entire period. Horizontal black lines represent two standard deviations on both sides of zero in the time series. The blue shaded areas also represent 2 standard deviations on both sides of zero but based on the time series from *pre-industrial control* simulations. The horizontal black line is dashed if the variance of the simulation with time-varying forcing (black line) is significantly different (F-test, considering a 5% level) from the variance of the simulation with fixed forcing; if not, it is solid. For the CESM1 model the black curve is the median of the ten ensemble members, each shown in shades of grey (the closer the ensemble members are to the median, the darker they are), and the standard deviation is the average of the standard deviation of all individual ensemble members.

## 5.2 The stability of large-scale tele-connections

Since East African hydroclimate fluctuations in models are mostly driven by rainfall (see Section 4.2), only rainfall is considered here. The modern-day large-scale tele-connections have already been studied in Section 3.3. Hence, the period considered in this section is 850-1850 AD, which allows us to focus on the last millennium without the influence of anthropogenic forcing.

5    Annual mean large-scale tele-connections between simulated East African rainfall and tropical SSTs over the (pre-industrial portion of the) last millennium differ substantially among models. However, for rainfall over the Challa/Naivasha region, all GCMs except CCSM4 agree in simulating a dipole pattern over the Indian Ocean, with positive (negative) correlation in the western (eastern) half of the basin (Fig. 10a), i.e. consistent with the IOD pattern and its effect on East African precipitation. Interestingly, while CCSM4, CESM1 and MPI-ESM-P do not show much difference between last-millennium and recent

10   tele-connections, this dipole is not simulated by GISS-E2-R, IPSL-CM5A-LR and BCC-CSM1-1 with recent rainfall data



(1950-2000; Fig. 4a). The role of equatorial Pacific SSTs during the last millennium is less clear among models, with both positive and negative correlations depending on the model selected.

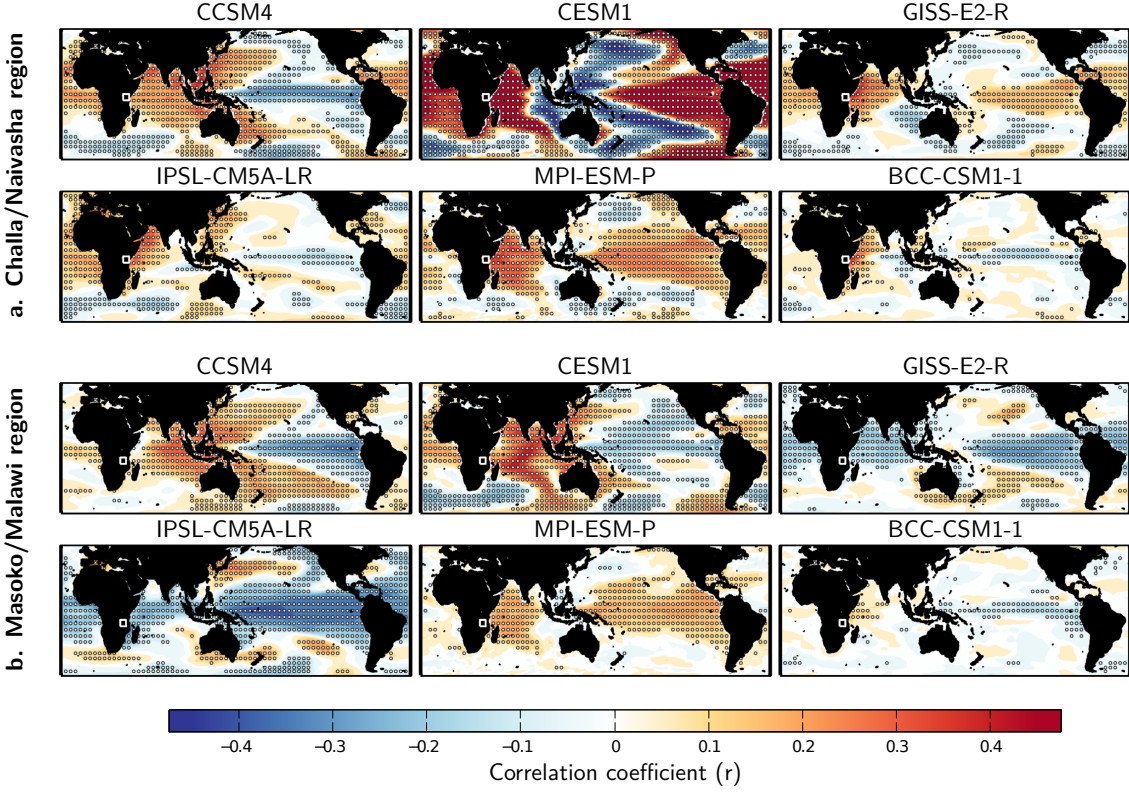

**Figure 10.** Pearson correlation coefficients between global SSTs and mean annual rainfall over the Challa/Naivasha (a) and Masoko/Malawi (b) regions, in climate models for the period 850-1850 AD. In areas overprinted by white circles the null hypothesis of no correlation can be rejected at the 5% level.

Patterns of correlation between simulated Masoko/Malawi rainfall and SSTs are more heterogeneous (Fig 10b). CCSM4 and MPI-ESM-P show a comparable picture as with Challa/Naivasha rainfall, which is also the case for CESM1 but only for the In-

5    dian Ocean. GISS-E2-R and, to a lesser extent, IPSL-CM5A-LR show an opposite pattern, and BCC-CSM1-1 suggests almost no link between Masoko/Malawi rainfall and SSTs. For this region, the pre-industrial last-millennium tele-connections differ strongly from the recent ones in all models except CCSM4 and GISS-E2-R (Fig. 4b). For this pre-industrial last-millennium period, the effect of changes in forcing on inter-annual rainfall tele-connections appears to be weak, given the very similar results of the last millennium runs (Fig. 10) and of the *pre-industrial control* runs for both regions (Fig. S3). It is thus likely

10   that the difference in tele-connection patterns between the pre-industrial period and recent decades is due to anthropogenic forcing. Based on the control runs of two global climate models, Tierney et al. (2013) showed that at longer than decadal time scales, rainfall over the Horn of Africa region (including our Challa/Naivasha domain) is mostly influenced by Indian Ocean SSTs. This is investigated here using smoothed GCM-simulation results with a window of 100 years. Such a smoothing



decreases drastically the number of degrees of freedom resulting in only a few statistically significant patterns of correlation in control simulations (Fig. 11). As regards simulated Challa/Naivasha rainfall, the correlation with SSTs displays the characteristic Indian Ocean dipole in all models except CCSM4. The dipole is especially clear and robust in CESM1, the model that best matches observations regarding recent large-scale tele-connections (Section 3.3). By contrast, at this centennial time scale,

5    the link between Masoko/Malawi rainfall and the Indian Ocean is neither consistent among models nor robust in these control simulations. Additionally, for neither Challa/Naivasha or Masoko/Malawi a significant link is obtained with Pacific SSTs.

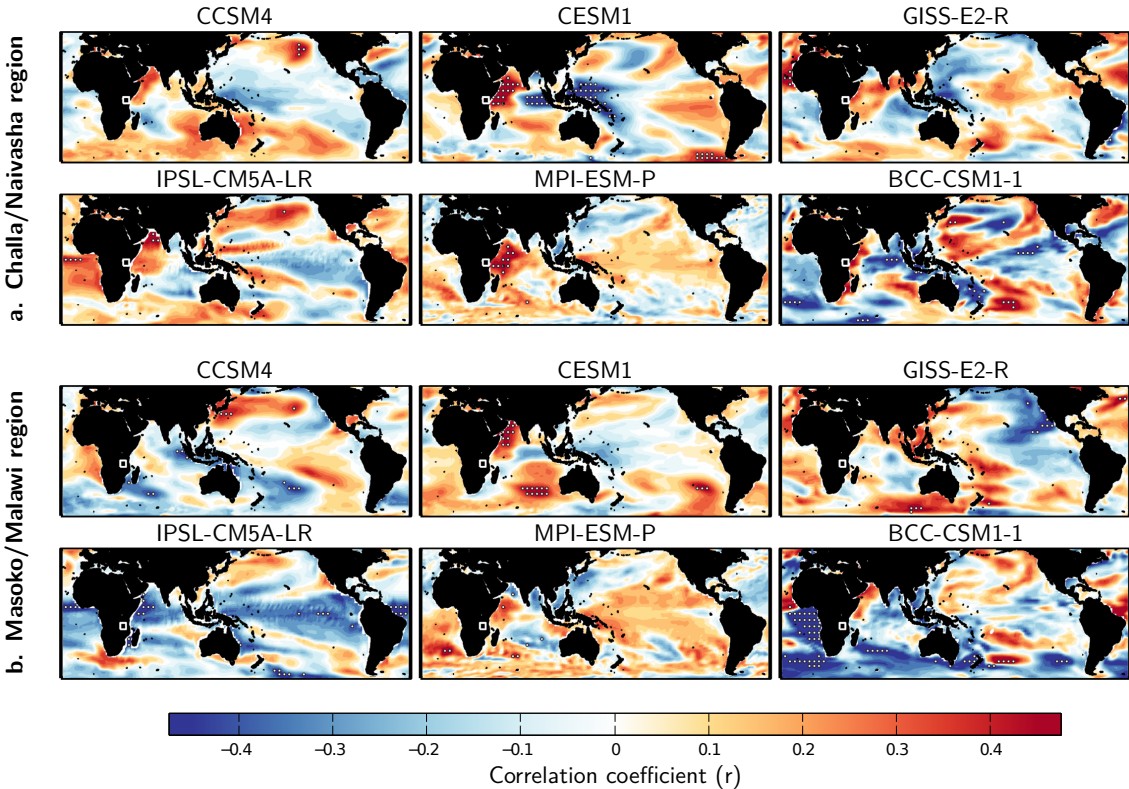

**Figure 11.** Pearson correlation coefficients between global SSTs and mean annual rainfall over the Challa/Naivasha (a) and Masoko/Malawi (b) regions, using the *pre-industrial control runs* of the climate models. Rainfall and SSTs are mean annual values smoothed using a loess filter with a window of 100 years. In areas overprinted by white circles the null hypothesis of no correlation can be rejected at the 5% level.

The patterns of correlation become completely different when considering last millennium simulations with changes in forcing (Fig. 12). Most correlations are not significant and relatively weak. Interestingly, in the forced runs no Indian Ocean dipole is observed in most model results at this centennial time scale and, more generally, there is substantial difference in

10    modeled large-scale tele-connections between simulations with time-varying forcing (Fig. 12) or fixed forcing (Fig. 11). This suggests that, at the centennial time scale, the forcing is able to mask the weak correlations associated with natural variability throughout the pre-industrial portion of the last millennium (850-1850 AD). Nevertheless, the response of the different models





to the same forcing strongly varies. Some models are characterized by a very homogeneous pattern of correlation that can be both negative or positive, while some models show a more patchy pattern.

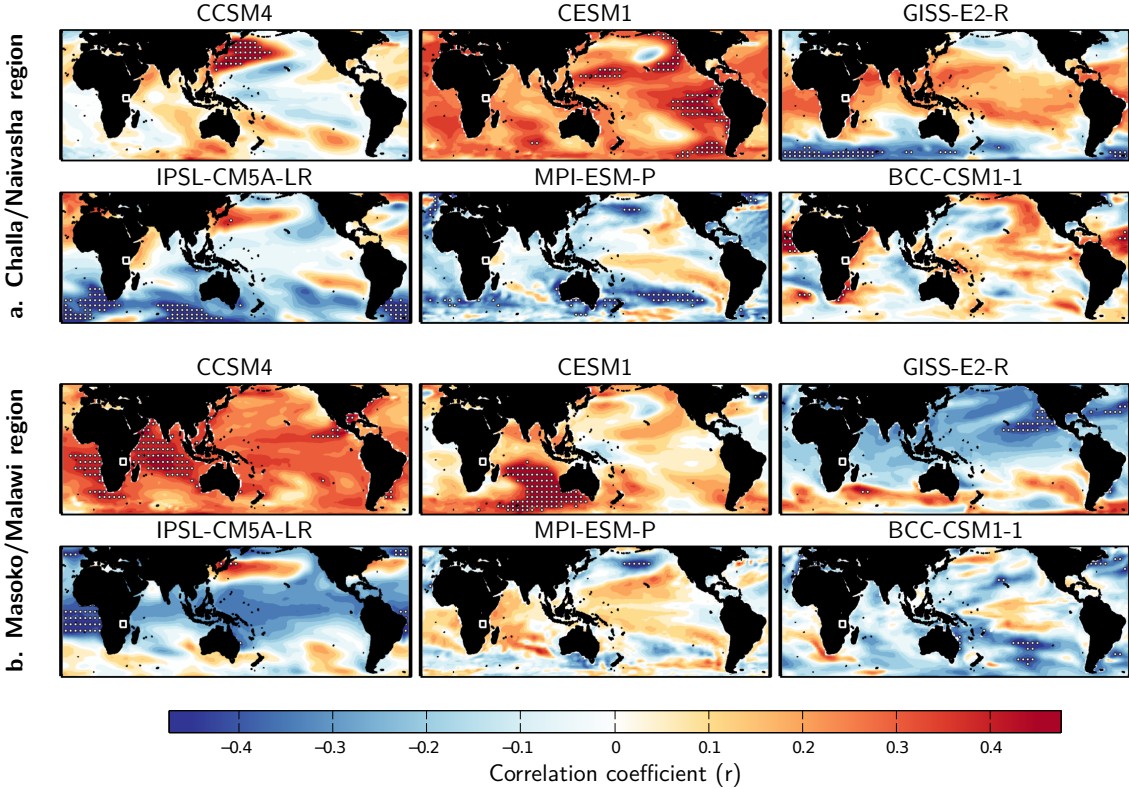

**Figure 12.** Pearson correlation coefficients between global SSTs and mean annual rainfall over the Challa/Naivasha (a) and Masoko/Malawi (b) regions in climate models for the pre-industrial period 850-1850 AD. Rainfall and SSTs are mean annual values smoothed using a loess filter with a window of 100 years. In areas overprinted by white circles the null hypothesis of no correlation can be rejected at the 5% level.

## 6   General discussion and conclusions

Our analysis of East Africa's hydroclimate over the last millennium is based on a comparison of recent observational data and lake-based proxy reconstructions with the results of six GCMs. When compared to recent observations, the GCM simulations represent the unimodal seasonality of precipitation characterizing the Masoko/Malawi spatial domain fairly well, and also the bimodal seasonality characterizing the Challa/Naivasha domain except that the relative magnitude of the two rainy seasons is less well captured. Model skill in simulating modern-day (i.e., observed) large-scale tele-connections between East African precipitation and tropical SSTs strongly vary among the GCMs, with MPI-ESM-P and CESM1 generally displaying the most consistent patterns.





Both model results and observations show that lakes Challa and Naivasha on the one hand, and Masoko and Malawi on the other are located in hydroclimatically relatively homogeneous regions. However, these two regions display a different rainfall seasonality and different large-scale tele-connections with ocean SSTs. Furthermore the lake-based proxy reconstructions from these two regions even show opposite moisture-balance changes during the second half of the last millennium, highlighting

the strong spatial heterogeneity characterizing East African hydroclimate dynamics. Comparing the simulated variable P-E with the available reconstructions, we found the contribution of rainfall to be dominant relative to actual evaporation in both regions. Model results and reconstructions show a very different timing of hydroclimate fluctuations over the past millennium. Furthermore, there is no common signal among the time series modeled by different GCMs. This suggests that simulated P-E in East Africa is largely driven by internal variability rather than by common forcing, at least until 1850 AD. After that, half

of the used GCMs simulate a relatively clear, but model-specific, response to forcing. These results are in line with those of Coats et al. (2015) who showed, using approximately the same set of GCMs, that multi-decadal droughts in the North American Southwest over the last millennium do not seem to be driven by external forcing. Similar conclusions were also reached by Kelley et al. (2012) who used GCMs to investigate the possibility that the late winter drying trend observed in the Mediterranean region could be explained by anthropogenic forcing. In contrast, Fallah and Cubasch (2015) suggested an

impact of forcing on multi-decadal droughts in Asia over the last millennium, namely through alteration of atmosphere-ocean interactions. These contrasting results could mean that some regions are more sensitive to forcing than others.

At the inter-annual time scale, models show robust tele-connections between mean annual Indian Ocean SSTs and rainfall over the Challa/Naivasha region during the pre-industrial portion of the last millennium, with positive (negative) correlation in the western (eastern) half of the basin. The link between rainfall over the Masoko/Malawi region and SSTs is less clear among

models. At this time scale, the effect of external forcing on large-scale tele-connections appears negligible. Although most of times not significant, the Indian Ocean dipole is still present using time series smoothed to highlight centennial variations, but only in fixed-forcing simulations. When taking into account the last millennium forcing, the result is completely different, with relatively homogeneous patterns of correlation between precipitation in both regions and tropical SSTs. This means that, although the correlation pattern between Challa/Naivasha rainfall and Indian Ocean SSTs remains relatively similar for both

inter-annual and centennial time scales when only natural variability is present, it is overwhelmed by the effect of forcing at the centennial time scale. An interesting question is whether the forcing actually alters the dynamical link between East African rainfall and SSTs, or if it only masks it because of a different impact on continental rainfall and SSTs. Answering this question is out of the scope of this study, but it is of interest for the interpretation of records used for reconstructing phenomena like the IOD. Indeed, if dynamical relationships are not stable when considering different time scales, a record calibrated in

observations of the recent period may not be representative of the studied phenomena over longtime scales.

Analyzing an ensemble of models was particularly useful here to test the robustness of the results. Additionally, using individual models that contain several ensemble members allows attributing the simulated change to internal variability or to forcing, which is crucial in the present context of climate change in the vulnerable East African region. On the other hand, using multi-model mean results to study variation in local hydroclimate during the last millennium should be avoided. It does



not make sense to average hydroclimate time series which are mostly driven by internal variability, since this would only result in concealing already weak responses.

*Acknowledgements.* This work is supported by the BRAIN-be programme of the Belgian Federal Science Policy Office (BelSPO) through project BR/121/A2 'Patterns and mechanisms of climate extremes in East Africa' (PAMEXEA). We acknowledge the World Climate Re-
5 search Programme's Working Group on Coupled Modelling, which is responsible for CMIP, and we thank the climate modeling groups (listed in Table 1 of this paper) for producing and making available their model output. For CMIP the U.S. Department of Energy's Program for Climate Model Diagnosis and Intercomparison provides coordinating support and led development of software infrastructure in partnership with the Global Organization for Earth System Science Portals. Hugues Goosse is research director with the FRS/FNRS, Belgium.

+ Flavio for CESM output.



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
