# Peer review of "Comparison of simulated and reconstructed variations in East African hydroclimate over the last millennium"

_Climate of the Past, 2015_

## Short Comment (SC1) · 19 Feb 2016

Sebastian Luening

sebastian.luning@gmx.net

This paper deals with a very important subject: Comparison of the simulated and reconstructed climate. Working out similarities and differences is key for a better understanding of the climate drivers and their quantification.

The authors have chosen four fairly high-resolution climate curves from East Africa which they compare with model results. Interestingly, some of the reconstructed climate curves differ markedly. In Figure 5 the Naivasha and Masoko lakes show a dry Medieval Warm Period (MWP) / Medieval Climate Anomaly. In contrast, Challa appears to be more humid, even though the record starts slightly later and the beginning is unclear. In the Lake Malawi climate curve the time 1000-1300 AD is ab-

sent, therefore it is unclear if the MWP was dry or wet here. I suggest you add information from Johnson et al. 2004. According to those authors: "Diatom productivity was high during the Little Ice Age (LIA) and relatively low around 1 kyr, the time of the Medieval Warm Period (MWP)". The low diatom productivity during the MWP may be linked to low river discharge, i.e. drought conditions. During this time the rivers may have supplied lower amounts of dissolved silica to the lake. During the wetter Dark Ages Cold Period and Little Ice Age, chemical weathering of bedrock intensified and increased the BSi concentrations and diatom productivity in the lake. http://link.springer.com/chapter/10.1007

I would like to draw your attention to an ongoing project in which I am mapping the climate characteristics of the MWP on a global scale, based on the large number of published case studies. The interactive online map is freely accessible here: http://t1p.de/mwp

In East Africa you see a large number of yellow points that represent studies which reported drought/arid conditions for the MWP time. When you click on the respective dot, key information from the paper appears, including a link to the key climate curve. Arid conditions seem to be the general pattern that existed 1000-1300 AD in East Africa.

The arid MWP belt appears to continue northwards along the coast of the Arabian Sea, including Ethiopia, Yemen, Oman, Pakistan and coastal northwestern India. There, the MWP climate regime seems to change. Southern and eastern India and the Bay of Bengal appear to be humid during the MWP. Mapping is still ongoing and many more studies will have to be integrated. It is also clear that in detail things are more complex. Nevertheless, I think it would be important to initially compare the models to these general, high-level patterns.

From your study and reference list I have gathered quite a few new publications that I will add to the MWP map in due course. Thanks for that.

Concerning the forcing of pre-industrial climate change, I am not comfortable with models that gain their simulated climate variability mostly from internal variability. There are clear MWP patterns and additional millenniums cycles (e.g. Bond et al. 2001) which point towards powerful external climate drivers. Numerous papers have highlighted the important role that solar activity changes play in the climate equation. I want to encourage you to also run models and scenarios with a solar radiative forcing higher than that assumed by the IPCC. If not for this paper, maybe in a future one. The current RF proposed by the IPCC does not honour the great number of studies which highlighted the intense coupling of climate with solar activity changes: http://chrono.qub.ac.uk/blaauw/cds.html

---

## Referee Comment (RC1) · Anonymous Referee #1 · 11 Mar 2016

The manuscript presents a comparative analysis of proxy records representative of hydroclimate in Eastern Africa and corresponding time series from climate simulations over the past millennium. After discussing the caveats due model spatial resolution and spatial homogeneity of precipitation this region, that authors reach the main conclusion that most of the hydroclimate variability in this region is probably caused by internal process and that the influence of external forcing seems to be very limited, in agreement with other studies that have pointed out the importance of internal variability in hydroclimate other parts of the world. Another important conclusion is that the different models do not agree in simulating the links between hydroclimate and sea-surface-temperatures. I think the research question is important and opens up further questions, as for instance the reasons why models diverge when simulating the SST-hydroclimate link which also leads to the question of the origin of hydroclimate variability itself and its connections to global patterns of climate variations like ENSO or the Indian Ocean dipole. My general impression of the manuscript is quite positive. The manuscript is rather long and, although at some stages the study falls short of reaching robust conclusions, I think it is a worthwhile contribution and opens up some lines of research for further studies. I liked the amount of manuscript space devoted to check the spatial representativity of the hydroclimate records, the skill of the models in simulating the two different precipitation annual cycles and the teleconnections to the large-scale SSTs, although I have a comment on this last point.

I have some comments on the manuscript that the authors may want to consider. Only two of them are general enough to possibly require some major changes in the manuscript, the rest being more more specific.

-I would like to start, however, underlying that the submitted version does not appear to have been thoroughly revised by the authors. Something seems to have gone awry regarding the blank spaces to separate words, and may words throughout the manuscript appear juxtaposed , at least in the pdf copy I downloaded. This has made the reading quite uncomfortable. This impression is confirmed by the acknowledgements to Flavio (?). I believe it is appropriate to acknowledge him by his full name.

-My main concern is the claim that precipitation, relative to evaporation, is the main factor driving hydroclimate variability. The authors compare the standard deviation of precipitation and evaporation in the model output and reach the conclusion that the former is much larger, with a few exceptions in the Challa/Naivasha region. However, this calculation is done at interannual timescales, as far as I can judge comparing the much larger magnitude of the standard deviations shown in Figure 6 than those shown in Figure 8, which are explicitly calculated at centennial timescales. If this is correct, I think this conclusion could be premature, since at longer timescales the variability of temperature would likely grow relative to the variability of precipitation, and thus also

the role of evaporation could become more important. I think this should be checked because the authors base some of the further analysis on this conclusion, and because it is a quite relevant conclusion on its own right.

-Another point is related to the teleconnections between precipitation and sea-surface-temperature described in section 3.3. This section directly assumes that the SST is a direct driver of precipitation, but the text does not contain a justification for this assumption. Could it be that both SSTs and precipitation are driven by the atmospheric circulation? In this region, both may be coupled being part of some coupled mode of variability, but it is also possible that the atmosphere is driving both. This possibility is related to the main conclusion of the paper that the influence of external forcing is negligible, as the atmosphere circulation would be arguably less responsive to forcing than the SST.

Particular points:

-The 1000-year time series representing hydroclimate variation in the Lake Challa region in Tierney et al. (2013) is the first principal component of composite variation in three moisture-balance proxies, namely a presumed indicator of catchment.

It would be useful to quote the variance explained by this leading PC. Is it clearly over 30%, which would be the expected value if the three series were uncorrelated ?

-pattern during an El Niase of ENSO

typo

-the series has been linearly standardized so that the maximum of theabsolute values equals 1.

This standardization is not really robust, as it depends on one single value: the maximum element in the series. The amplitude of the standardized series may therefore depend on an outlier.

-I really had to wrestle to understand Figure 9. First, I could not see the individual simulations of the CESM ensemble in panel upper row centre, apparently drawn with different shades of grey according to their distance to the median. I do not think it is necessary to show the distance to the median (what is the reason ?), and it quite messes up the figure. What would be the distance criterion anyway ? Second, I did not understand the blue shading. It apparently shows the 2x standard deviation derived boundaries from a control run, added to the median simulation (?). But it seems that the blue shading is not simply the median line with (constant) 2xsigma boundaries added. The blue-shaded area has a time evolution that is different to that of the median simulation. Third, if the blue-shading indicates the standard deviation from control simulations, it is much smaller that the standard deviation from the forced simulation , so why is the line indicating the latter dashed? My impression is that this caption is not quite right. Maybe the blue shading indicates the within-ensemble standard deviation from the CESM ensemble (and so it is time-evolving as well) and not the standard deviation from a control run (constant). I think I see the point that the authors are trying to make in this panel. Perhaps the authors may want to consider just showing one or two simulations from the CESM ensemble if what they wish is to convey the amplitude of variations as compared to the other models. Showing the median is misleading (compared to other models), and showing the within-ensemble standard deviation (if this is what the blue-shading indicates) does not alleviate the problem.

---

## Referee Comment (RC2) · Anonymous Referee #2 · 25 Mar 2016

Summary: This study uses a paleoclimate model data comparison framework to analyze East African lake levels over the last millennium. GCMs struggle to represent the seasonal cycle of precipitation and teleconnections over East Africa. Nevertheless, the telconnections appear to be variable over the last millennium, and between fixed forcing and variable forcing simulations. For the Masoko/Malawi region, in particular, anthropogenic forcing appears to influence the teleconnections. On centennial timescales the variation in telelconnections are large for both regions and this is explained by changes to natural forcing. Despite a clear link between forcing and teleconnection changes in the models over the last millennium, there is no relationship between forcing and hydroclimate changes. By contrast, internal atmosphere-ocean variability is shown to be

Interactive
comment

the dominant driver of simulated hydroclimate changes over East Africa, even on centennial timescales (although anthropogenic has driven consitent simulated changes in the Masoko/Malawi region over the most recent ∼150 years). The dominant role for internal atmosphere-ocean variability in driving hydroclimate changes can explain the mismatch between the time histories of hydroclimate over East Africa simulated by models and that reconstructed for the four lakes.

General Remarks: This represents an interesting and important contribution to our understanding of low-frequency hydroclimate variability over East Africa as simulated by models. My comments are largely minor although I do have three major concerns that I hope that the authors will consider addressing.

Major Concerns:

1. The manuscript, in general, is clear although I would strongly suggest revisiting the manuscript and editing for grammar and sentence structure.

2. (Page 23, Line 6) "An interesting question is whether the forcing actually alters the dynamical link between East African rainfall and SSTs, or if it only masks it because of a different impact on continental rainfall and SSTs. Answering this question is out of the scope of this study, but it is of interest for the interpretation of records used for reconstructing phenomena like the IOD. Indeed, if dynamical relationships are not stable when considering different time scales, a record calibrated in observations of the recent period may not be representative of the studied phenomena over longtime scales." This is very important and likely the most important conclusion of the paper, is there not some relatively simple way to approach this (specifically determining if the forcing actually alters the dynamical link between East African rainfall and SSTs, or if it only masks it because of a different impact on continental rainfall and SSTs)? I think it would really improve the contribution that this paper will make to our understanding of simulated and real-world East African rainfall. In particular, the CESM last millennium ensemble would seem to be well suited to answering this question given that the ten

ensemble members can be used to robustly determine the impacts of forcing relative to internal variability.

3. When analyzing the reconstructions, Challa and Naivasha look very different, as do Malawi and Masoko. I found that the descriptions of common changes here were not consistent with what can be seen by eye in the figure. Perhaps there is a more quantitative way to approach this? Generally, I suggest that this section be revisited. If the reconstructions do not line up why might this be and what doest this suggest for our interpretation of the model simulations?

Specific Comments:

Abstract: "The bimodal seasonal cycle characterizing the Challa/Naivasha region, except that in the latter the relative magnitude of the two rainy seasons is less well captured." This language doesn't seem fully consistent with the results, it appears that the models generally struggle to reproduce the characteristics of the seasonal cycle at both locations.

Introduction: Southeast and equatorial east African lakes versus east African rainfall. Make as clear as possible at the outset that east African is covering all four lakes but that the distinction between southeast and equatorial east African is how you will describe the two sets of two lakes.

(Page 2, Line 8) "through atmospheric adjustments to the Walker circulation". This statement is unclear to me, perhaps you mean oceanic driven changes to the atmospheric Walker circulation?

(Page 2, Line 23) "Enhanced pattern" perhaps better to say an increase in because you are not talking about a spatial pattern.

(Page 2, Line 33) "poor ability" suggest using the inability.

(Page 3, Line 1) "reached contrasting conclusions depending on the region or spatial scale, or on the variables and models considered." Given this, basing any conclusions

regarding the role of internal variability on the fact that the models do not match the reconstructions seems problematic given that only two model grid points are being analyzed. A larger spatial scale might provide more confidence here, however the results using the CESM ensemble do provide strong evidence for the role of internal variability.

(Page 3, Line 20) It is not clear how the annual means follow from the long-term changes. Maybe save the discussion regarding the use of annual means portion for later.

(Page 3, Line 22) Is there any reason to suggest that models should capture the reconstructed changes? Perhaps change to analyze GCM simulations of long-term change relative to reconstructions over the last millennium. I do not think this study is truly aimed at investigating GCM performance.

(Page 4) An aside, but I really appreciate the detail that you have gone into with regards to the set up of the models and the slight differences.

Section 2.2. I think it is important to note how comparable each of these records are. Are there reasons to expect systematic differences given that they each are reconstructing different things? How might this impact the interpretation of the results and conclusions?

(Figure 2) On the x-axis Fev should be Feb, the labels overlap so potentially make text smaller.

(Page 7, Line 15) To my eye CESM1 doesn't look better than the other models, BCC is arguably more realistic, maybe just remove the second part of that sentence.

(Page 8, Line 1) Put a space in "Lake Naivashais".

(Page 8, Line 4) I am not sure the split in how realistic the models are is that clear. When looking at CCSM4, CESM1 and BCC-CSM1-1, IPSL looks just as reasonable to me. Maybe be a bit more general.

(Page 8, Line 13) I think this is a bit of an oversimplification in the second part of this sentence as the timing is also off (not just the magnitude).

(Page 8, Line 22) "Since Lake Challa and Lake Naivasha on the one hand and Lake Masoko and Lake Malawi on the other have a very similar climatology and seasonal cycle both in models and observations (Fig. 2)". We are interested in long term changes, nothing that can really be done but the reconstructions for each lake look very different suggesting that on long timescales things might not be expected to be similar.

(Page 8, Line 24) "However, using the larger grid boxes raises the issue of whether the proxy-based reconstructions employed to assess model performance through the last millennium are representative for these larger spatial domains". Be more exact here, don't use assess model performance—that's not really what's being done here.

(Page 9, Line 12) "Regarding annual mean absolute values". What do you mean by this?

(Page 10, Line 34) "Nevertheless, with values equal to 0.27 and 0.16 the significant correlation coefficients between, respectively." Are these spatial correlations? This section is a bit confusing; suggest revisiting with an eye towards clarifying the language.

(Page 12, Line 19) "theright", put space.

(Page 12, Line 27) "Making abstraction of chronological uncertainties, the Challa and Naivasha proxy records show discrepancy during the first four centuries of the last millennium". This sentence is confusing.

(Page 12, Line 30) "humidity", suggest changing to wetting.

(Page 13, Line 7) "Besides", suggest removing.

(Page 14, Line 2) "thus leads to larger river runoff towards the lakes". I am not sure that I understand this as the models do not have these actual lakes.

Page 14, Line 5) "This implies that most changes of P-E over time are due to changes

in precipitation". Is this necessarily true on longer timescales?

(Page 15, Line 8) Why not a more typical and interpretable standardization using the standard deviation?

(Page 15, Line 9) How does this or does this not match the temporal resolution of those reconstructions. I suppose I am just interested in a more thorough justification for this choice of smoothing.

(Page 15, Line 11) By eye in Figure 7 the model variability looks to be about the same magnitude as the reconstructions.

(Page 16, Line 8) "Spatial representivity", should likely just be locations.

(Page 16, Line 12) "confronted", should be compared.

(Page 17, Line 10) This is an important sentence. At least remove "that" to make it clear, however, restructuring the sentence would probably be good.

(Page 17, Line 22) "In this regard, it is of interest to note that for the one model for which multiple ensemble members were available (CESM1), there is also no correlation between the different ensemble members that differ only from slightly different air temperature at the start of the experiments (Otto-Bliesner et al., 2015)". This is very important and gets lost a bit as cast.

(Section 5.2) I feel like this section would be clearer with the unfiltered pre-industrial control run teleconnections also shown. It becomes a bit confusing with the pre-industrial portion of the last millennium, pre-industrial control runs, and historical simulations all being compared simultaneously.

(Conclusion) This ends on a weak note, I might suggest finishing the paper with the last sentence of the previous paragraph.

---

## Author Comment (AC1) · 10 May 2016

- In blue: referees' comments

- In black: our answers

- *In black italic: what we propose to add in the text*

The manuscript presents a comparative analysis of proxy records representative of hydroclimate in Eastern Africa and corresponding time series from climate simulations over the past millennium. After discussing the caveats due model spatial resolution and spatial homogeneity of precipitation this region, that authors reach the main conclusion that most of the hydroclimate variability in this region is probably caused by

internal process and that the influence of external forcing seems to be very limited, in agreement with other studies that have pointed out the importance of internal variability in hydroclimate other parts of the world. Another important conclusion is that the different models do not agree in simulating the links between hydroclimate and sea-surface-temperatures. I think the research question is important and opens up further questions, as for instance the reasons why models diverge when simulating the SST-hydroclimate link which also leads to the question of the origin of hydroclimate variability itself and its connections to global patterns of climate variations like ENSO or the Indian Ocean dipole. My general impression of the manuscript is quite positive. The manuscript is rather long and, although at some stages the study falls short of reaching robust conclusions, I think it is a worthwhile contribution and opens up some lines of research for further studies. I liked the amount of manuscript space devoted to check the spatial representativity of the hydroclimate records, the skill of the models in simulating the two different precipitation annual cycles and the teleconnections to the large-scale SSTs, although I have a comment on this last point.

I have some comments on the manuscript that the authors may want to consider. Only two of them are general enough to possibly require some major changes in the manuscript, the rest being more more specific.

- I would like to start, however, underlying that the submitted version does not appear to have been thoroughly revised by the authors. Something seems to have gone awry regarding the blank spaces to separate words, and may words throughout the manuscript appear juxtaposed, at least in the pdf copy I downloaded. This has made the reading quite uncomfortable. This impression is confirmed by the acknowledgements to Flavio (?). I believe it is appropriate to acknowledge him by his full name.

We would like to apologize for that. Something went indeed awry when compiling our manuscript with the standard LaTeX template of Climate of the Past. We have noticed that the day after the submission and have then sent a corrected version of the manuscript, but it took some time to be updated. Anyway, we should have more

thoroughly checked the initially submitted manuscript and we are sorry for the inconvenience.

- My main concern is the claim that precipitation, relative to evaporation, is the main factor driving hydroclimate variability. The authors compare the standard deviation of precipitation and evaporation in the model output and reach the conclusion that the former is much larger, with a few exceptions in the Challa/Naivasha region. However, this calculation is done at interannual timescales, as far as I can judge comparing the much larger magnitude of the standard deviations shown in Figure 6 than those shown in Figure 8, which are explicitly calculated at centennial timescales. If this is correct, I think this conclusion could be premature, since at longer timescales the variability of temperature would likely grow relative to the variability of precipitation, and thus also the role of evaporation could become more important. I think this should be checked because the authors base some of the further analysis on this conclusion, and because it is a quite relevant conclusion on its own right.

Thank you for pointing this out. You are right, in the submitted manuscript the comparison between the simulated standard deviation of precipitation and evaporation is computed using annually averaged results for the last millennium (850-2005 AD; Fig. 6 of the manuscript). One could indeed expect an increasing importance of temperature and thus of evaporation when using the results averaged over longer timescales. However, this is not the case in the models (Fig. 1 of this document). Using a smoothing window of 100 years, the differences between the standard deviation of precipitation and evaporation drop by a factor of about 10, but the relative amplitudes remain approximately the same to what is shown using annual results, meaning that precipitation still dominates on evaporation.

We propose to add the following sentence at the end of Section 4.2:

*Moreover, also at timescales longer than annual, rainfall has a dominant role in explaining changes in P-E, since approximately the same picture is observed when smoothing*

*model results with a loess filter window of 100 years (not shown).*

- Another point is related to the teleconnections between precipitation and sea-surface-temperature described in section 3.3. This section directly assumes that the SST is a direct driver of precipitation, but the text does not contain a justification for this assumption. Could it be that both SSTs and precipitation are driven by the atmospheric circulation? In this region, both may be coupled being part of some coupled mode of variability, but it is also possible that the atmosphere is driving both. This possibility is related to the main conclusion of the paper that the influence of external forcing is negligible, as the atmosphere circulation would be arguably less responsive to forcing than the SST.

You are right. Although the association between tropical Indian and Pacific Ocean SSTs and East African rainfall has been emphasized in numerous studies (see references in the introduction), only a few of them really focus on understanding the underlying mechanisms. Here, we observe that SSTs and precipitation are correlated. However, based on available climate model experiments we cannot say whether both variables are part of a coupled mode of variability, are dynamically linked independently of a mode, or are just correlated without any dynamical link. Answering this question would probably require additional simulations with sensitivity experiments, which is outside the scope of this study.

Nevertheless, it is instructive to test if models are able to reproduce observed correlations, whatever the cause of those correlations. We propose to explain that more explicitly in Section 3.3 of the revised version of the manuscript:

*In addition to evaluation of the mean state, it is important to assess the ability of climate models to represent the observed regional patterns in the inter-annual variability of East African rainfall. Although it is not our goal here to study the mechanisms responsible for this simulated variability, it is illustrated by calculating the correlation between East African precipitation and tropical SSTs, which are considered a direct*

*driver of precipitation over East Africa (e.g. Goddard and Graham, 1999; Ummenhofer et al., 2009).*

We will also clearly state in Section 3.3 that these correlations do not necessarily imply dynamical links.

Particular points:

- The 1000-year time series representing hydroclimate variation in the Lake Challa region in Tierney et al. (2013) is the first principal component of composite variation in three moisture-balance proxies, namely a presumed indicator of catchment [precipitation]. It would be useful to quote the variance explained by this leading PC. Is it clearly over 30%, which would be the expected value if the three series were uncorrelated?

PC1 accounts for 40% of the variance in the 3 data time series, as mentioned in the supplementary materials of Tierney et al. (2013). This information will be added in Section 2.2 (Proxy-based hydroclimate reconstructions).

- pattern during an El Niase of ENSO. typo

This will be corrected.

- the series has been linearly standardized so that the maximum of the absolute values equals 1. This standardization is not really robust, as it depends on one single value: the maximum element in the series. The amplitude of the standardized series may therefore depend on an outlier.

Here, the outliers do not have an important impact on the amplitude of the standardized series since the raw time series are annually averaged and filtered before being standardized. We limited the amplitude of the time series between -1 and 1 mainly for aesthetic reasons. If we standardize by dividing the time series by their standard deviation instead of their maximum, the resulting figure is qualitatively the same (Fig. 2 of this document). We propose thus to keep the figure as in the submitted manuscript.

Interactive
comment

- I really had to wrestle to understand Figure 9. First, I could not see the individual simulations of the CESM ensemble in panel upper row centre, apparently drawn with different shades of grey according to their distance to the median. I do not think it is necessary to show the distance to the median (what is the reason ?), and it quite messes up the figure. What would be the distance criterion anyway?

We wanted to show the information contained in all 10 ensemble members of CESM1 while having a clear picture. Showing all individual members makes the figure quite noisy (Fig. 3 a of this document), so our compromise was to draw the median of these ensemble members together with the range (Fig. 3 b of this document). The different shades of grey correspond to the number of ensemble members included between the median and the edges of the shades: for a given year, the furthest ensemble member (corresponding to the range) is thus drawn in the lightest grey, the second furthest in a bit darker grey, etc. We are sorry you could not discern between the different shades of grey and will make sure that the distinction between them is adequate in the revised version of this figure.

Second, I did not understand the blue shading. It apparently shows the $2\times$ standard deviation derived boundaries from a control run, added to the median simulation (?). But it seems that the blue shading is not simply the median line with (constant) $2\times$sigma boundaries added. The blue-shaded area has a time evolution that is different to that of the median simulation.

The blue shading is indeed $2 \times$ standard deviation computed from the pre-industrial control runs, but around zero, as mentioned in the caption. The shading is thus drawn between $-2 \times$ standard deviation to $+2 \times$ standard deviation. It is not added to the median simulation. These values are constant through time.

Third, if the blue-shading indicates the standard deviation from control simulations, it is much smaller than the standard deviation from the forced simulation, so why is the line indicating the latter dashed?

If you refer to the results from CESM1 in the Masoko/Malawi region, the standard deviation of the control run is indeed smaller than the standard deviation of the forced simulation (which is not the case for Challa/Naivasha). It has to be noted that in the case of CESM1, the standard deviation of the forced run shown is the mean of the standard deviation of each ensemble member, and not of the standard deviation of the median which would have prevented any valuable comparison with the standard deviation of the single control run.

The red line, which represents the mean of $2\times$ standard deviation of each forced run of CESM1 around zero, is dashed because it is significantly different from the same diagnostic computed from the control run, according to a F-test (5% level).

My impression is that this caption is not quite right. Maybe the blue shading indicates the within-ensemble standard deviation from the CESM ensemble (and so it is time-evolving as well) and not the standard deviation from a control run (constant). I think I see the point that the authors are trying to make in this panel. Perhaps the authors may want to consider just showing one or two simulations from the CESM ensemble if what they wish is to convey the amplitude of variations as compared to the other models. Showing the median is misleading (compared to other models), and showing the within-ensemble standard deviation (if this is what the blue-shading indicates) does not alleviate the problem.

Given the above information, we don't think that the caption is wrong, but we propose to modify it to be more explicit:

**Figure 9:** *Simulated time series of P-E over the Challa/Naivasha and Masoko/Malawi regions throughout the last millennium (850-2005). Results are mean annual values smoothed using a loess filter with a window of 100 years, and are presented as anomalies with respect to the entire period. The horizontal red lines are displaced on both sides of the zero line at two times standard deviation of the smoothed time series. The horizontal blue lines also represent 2 standard deviations on both sides of zero*

*but based on the time series from pre-industrial control simulations. The horizontal lines are dashed if the variance of the simulation with time-varying forcing (black line) is significantly different (F-test, considering a 5% level) from the variance of the simulation with fixed forcing; if not, it is solid. For the CESM1 model, the black curve is the median of the ten ensemble members, while the range is shown in grey shading. Within the grey area, the ranges excluding the furthest ensemble members above and below the median, the furthest two ensemble members above and below the median and the furthest four ensemble members above and below the median are drawn using increasingly darker shades of grey.*

Also, to make the figure clearer, we propose to replace the blue-shaded area by two horizontal blue lines only, and to change the colour of the horizontal black lines to red, as in the Fig. 4 of this document.

———————————————

[Figure]

[Figure]

**Figure 1** – Mean annual precipitation minus mean annual evaporation (horizontal bars), and standard deviation of annual precipitation minus the standard deviation of annual evaporation (vertical bars) using smoothed results with a window of 100 years. The period considered is the entire last millennium (850-2005). The differences between the standard deviation of precipitation and evaporation (vertical bars) are multiplied by 10.

**Fig. 1.**

[Figure]

**Figure 2** – Comparison between last-millennium time series of the reconstructions (in grey) and of P-E simulated by six GCMs (in black) averaged over the Challa/Naivasha region (a), with the Naivasha record shown as dashed line and the Challa record as solid line; and over the Masoko/Malawi region (b), with the Malawi record shown as dashed line and the Masoko record as solid line. In both regions, the area between the two records is shaded in light grey. Both proxy-based and simulated time series are presented as anomalies with respect to the whole period, and are standardized by being divided by their standard deviation. Ordinate axes are oriented such that wetter (drier) conditions point upwards (downwards). Model time series are annual mean values filtered using a loess method with a window of 100 years. For the CESM1 model, the black curve is the median of the ten ensemble members previously standardized and smoothed.

**Fig. 2.**

[Figure]

**(a)** 10 ensemble members of CESM1 separated.

[Figure]

**(b)** Median of the 10 ensemble members and range.

Figure 3

**Fig. 3.**

[Figure]

**Figure 4** – Simulated time series of P-E over the Challa/Naivasha and Masoko/Malawi regions throughout the last millennium (850-2005). Results are mean annual values smoothed using a loess filter with a window of 100 years, and are presented as anomalies with respect to the entire period. The horizontal red lines are displaced on both sides of the zero line at two times standard deviation of the smoothed time series. The horizontal blue lines also represent 2 standard deviations on both sides of zero but based on the time series from pre-industrial control simulations. The horizontal lines are dashed if the variance of the simulation with time-varying forcing (black line) is significantly different (F-test, considering a 5% level) from the variance of the simulation with fixed forcing; if not, it is solid. For the CESM1 model, the black curve is the median of the ten ensemble members, while the range is shown in grey shading. Within the grey area, the ranges excluding the furthest ensemble members above and below the median, the furthest two ensemble members above and below the median and the furthest four ensemble members above and below the median are drawn using increasingly darker shades of grey.

**Fig. 4.**

---

## Author Comment (AC2) · 10 May 2016

- In blue: referees' comments
- · In black: our answers
- · In black italic: what we propose to add in the text

Summary: This study uses a paleoclimate model data comparison framework to analyze East African lake levels over the last millennium. GCMs struggle to represent the seasonal cycle of precipitation and teleconnections over East Africa. Nevertheless, the teleconnections appear to be variable over the last millennium, and between fixed forcing

and variable forcing simulations. For the Masoko/Malawi region, in particular, anthropogenic forcing appears to influence the teleconnections. On centennial timescales the variation in telelconnections are large for both regions and this is explained by changes to natural forcing. Despite a clear link between forcing and teleconnection changes in the models over the last millennium, there is no relationship between forcing and hydroclimate changes. By contrast, internal atmosphere-ocean variability is shown to be the dominant driver of simulated hydroclimate changes over East Africa, even on centennial timescales (although anthropogenic has driven consistent simulated changes in the Masoko/Malawi region over the most recent 150 years). The dominant role for internal atmosphere-ocean variability in driving hydroclimate changes can explain the mismatch between the time histories of hydroclimate over East Africa simulated by models and that reconstructed for the four lakes.

General Remarks: This represents an interesting and important contribution to our understanding of low-frequency hydroclimate variability over East Africa as simulated by models. My comments are largely minor although I do have three major concerns that I hope that the authors will consider addressing.

Major Concerns: 1. The manuscript, in general, is clear although I would strongly suggest revisiting the manuscript and editing for grammar and sentence structure.

We will check again the manuscript carefully and do our best to improve the grammar and sentence structure. It will eventually go through the copy editing process of Climate of the Past, which will help improve the language as well.

2. (Page 23, Line 6) "An interesting question is whether the forcing actually alters the dynamical link between East African rainfall and SSTs, or if it only masks it because of a different impact on continental rainfall and SSTs. Answering this question is out of the scope of this study, but it is of interest for the interpretation of records used for reconstructing phenomena like the IOD. Indeed, if dynamical relationships are not stable when considering different time scales, a record calibrated in observations of the
recent period may not be representative of the studied phenomena over longer time scales." This is very important and likely the most important conclusion of the paper, is there not some relatively simple way to approach this (specifically determining if the forcing actually alters the dynamical link between East African rainfall and SSTs, or if it only masks it because of a different impact on continental rainfall and SSTs)? I think it would really improve the contribution that this paper will make to our understanding of simulated and real-world East African rainfall. In particular, the CESM last millennium ensemble would seem to be well suited to answering this question given that the ten ensemble members can be used to robustly determine the impacts of forcing relative to internal variability.

Thank you for the suggestion, we agree that this is an important point. We can indeed determine the impact of the forcing on the teleconnections by comparing the average of the results of the 10 time-varying forcing experiments (r1-r10i1p1) with the results of the control simulation that fixes forcing to their pre-industrial values (Fig. 1 of this document).

We can see that the teleconnection patterns in the 'historical', 'past1000' and 'PI control' experiments of CESM1 are very similar. This is confirmed by the very small difference between 'historical' (simulation with the strongest forcing) and 'PI control' runs (lower row of Fig. 1 of this document). This means that the impact of forcing on teleconnection patterns is modest and that this pattern in CESM1 is dominated by internal variability. If we look in detail at the lower row of Fig. 1 of this document, we see that considering changes in forcing tends to dampen the teleconnections: the positive correlations get slightly less positive, and the negative correlations slightly less negative. This is interesting but the impact is, however, very small. Furthermore, this does not give any information about the mechanisms involved in the changes in the teleconnection patterns due to forcing, which would require new simulations with sensitivity experiments. Also, we can't consider these results as robust, since they are only based on a single model while most models show substantial differences in their
representation of the teleconnections studied. Hence, we think that the above analysis does not add value to the core of the manuscript, and prefer not to include it.

3. When analyzing the reconstructions, Challa and Naivasha look very different, as do Malawi and Masoko. I found that the descriptions of common changes here were not consistent with what can be seen by eye in the figure. Perhaps there is a more quantitative way to approach this? Generally, I suggest that this section be revisited. If the reconstructions do not line up why might this be and what does this suggest for our interpretation of the model simulations?

We agree with the referee: although common features do occur between Challa and Naivasha (peak wetting around 1700 AD) and between Masoko and Malawi (dry conditions around 1700 AD), these records are different. We propose to rewrite Section 4.1 accordingly:

Notwithstanding chronological uncertainty, the Challa and Naivasha proxy records display clear differences during the first four centuries of the last millennium. In particular, the former shows roughly a drying trend while the opposite is recorded in the latter (Fig. 5). However, from around 1400 AD the general trends inferred from these two records are similar: both show relatively dry conditions followed by a wetting trend peaking between about 1700 and 1750 AD. After this peak, both hydroclimate reconstructions depict an abrupt transition towards a dry period in the early 19th century, followed by smaller-scale hydroclimate fluctuations at Naivasha and a clear wetting trend at Challa.

<Figure 5 of the manuscript>

Contrasting with Challa and Naivasha, lakes Masoko and Malawi both show a general drying trend culminating around 1700 AD, before an increase in humidity towards the present. However, (multi-) decadal hydroclimate changes overlying these long-term trends often strongly differ from one another in the two records.

The differences between the Challa and Naivasha reconstructions on the one hand and
between the Masoko and Malawi reconstructions on the other can be viewed as a measure of the compound uncertainty of these proxy time series to represent the region's hydroclimate history. This may partly reflect real differences in the local hydroclimate history of these sites due to their different exposure to the principal seasonal moisture sources as affected by distance to the sea, topography etc. However, the most likely greatest source of time-series differences observed within each pair of records is due to the compound effects of i) dating uncertainty in these lake-based proxy records, ii) differences in hydrology and local catchment processes influencing a lake's (or its surrounding vegetation) sensitivity to climate, and/or iii) the fact that the used hydroclimate proxies have a specific and different relationship with temporal variation in our target of reconstruction, i.e. the climatic moisture balance. What is important here is that the differences between the two pairs of records are qualitatively more significant than those within each pair, to the extent that each pair is representative of a distinct hydroclimatic region (cf. Tierney et al., 2013). In the Challa/Naivasha region the main phase of the Little Ice Age equivalent period was wetter than average, and in the Masoko/Malawi region it was drier than average.

In the submitted manuscript, the simulated time series for Challa and Naivasha on the one hand and for Masoko and Malawi on the other are averaged over regions larger than the individual grid cells for those lakes. This is justified because according to climate models and recent observations (see Section 3.2) these sites are located in climatically homogeneous regions. In contrast, given the real differences between the reconstructions within each pair of sites, they are shown individually in our figures.

**Specific Comments:**

Abstract: "The bimodal seasonal cycle characterizing the Challa/Naivasha region, except that in the latter the relative magnitude of the two rainy seasons is less well captured." This language doesn't seem fully consistent with the results, it appears that the models generally struggle to reproduce the characteristics of the seasonal cycle at both locations.
We agree with the referee, this sentence appears too optimistic. Although some models actually perform quite well at representing the observed seasonal cycle in Challa and Naivasha (Fig. 2 of the submitted manuscript and Table 1 of this document), there is a large spread among models as mentioned in Section 3.1.

**Table 1.** Pearson correlation coefficients between the observed and simulated seasonal cycle over the period 1979-2005.

|          | CCSM4 | CESM1 | GISS-E2-R | IPSL-CSM5A-LR | MPI-ESM-P | BCC-CSM1-1 |
|----------|-------|-------|-----------|---------------|-----------|------------|
| Naivasha | 0.17  | 0.43  | 0.68      | 0.59          | 0.47      | 0.49       |
| Challa   | 0.37  | 0.65  | 0.57      | 0.92          | 0.50      | 0.70       |
| Masoko   | 0.90  | 0.96  | 0.87      | 0.89          | 0.83      | 0.80       |
| Malawi   | 0.95  | 0.97  | 0.90      | 0.92          | 0.88      | 0.82       |

In the abstract, we propose to replace:

"The GCMs simulate fairly well the unimodal seasonal cycle of precipitation in the Masoko/Malawi region and the bimodal seasonal cycle characterizing the Challa/Naivasha region, except that in the latter the relative magnitude of the two rainy seasons is less well captured."

**by**

All GCMs simulate fairly well the unimodal seasonal cycle of precipitation in the Masoko/Malawi region, while the bimodal seasonal cycle characterizing the Challa/Naivasha region is generally less well captured by most models.

In the conclusion, we propose to change:

"When compared to recent observations, the GCM simulations represent the unimodal seasonality of precipitation characterizing the Masoko/Malawi spatial domain fairly well, and also the bimodal seasonality characterizing the Challa/Naivasha domain except that the relative magnitude of the two rainy seasons is less well captured."
When compared to recent observations (1979-2005), simulations of all GCM models represent the unimodal seasonality of precipitation characterizing the Masoko/Malawi spatial domain rather well. The bimodal seasonality characterizing the Challa/Naivasha domain is generally less well captured by the models, with a systematic underestimation of the long rains and overestimation of the short rains.

Introduction: Southeast and equatorial east African lakes versus east African rainfall. Make as clear as possible at the outset that east African is covering all four lakes but that the distinction between southeast and equatorial east African is how you will describe the two sets of two lakes.

Following the referee's suggestion, we propose to change:

"In this study, we consider proxy records describing the water-balance history of Lake Challa and Lake Naivasha in eastern equatorial Africa, and of Lake Masoko and Lake Malawi in southeastern (but still inter-tropical) Africa (Fig. 1)."

**by**

In this study, we consider proxy records describing the water-balance history of four East African lakes: Lake Challa and Lake Naivasha in eastern equatorial Africa, and Lake Masoko and Lake Malawi in southeastern (but still inter-tropical) Africa (Fig. 1).

(Page 2, Line 8) "through atmospheric adjustments to the Walker circulation". This statement is unclear to me, perhaps you mean oceanic driven changes to the atmospheric Walker circulation?

It actually depends on the studies. The introduction will be modified in this way:

However, the mechanisms involved are less clear, with Goddard and Graham (1999) and Ummenhofer et al. (2009) suggesting that an ocean-driven change to the atmospheric Walker circulation impacts East African rainfall, while Klein et al. (1999)
mention an atmospheric change affecting both tropical-ocean SSTs and East African rainfall.

(Page 2, Line 23) "Enhanced pattern" perhaps better to say an increase in because you are not talking about a spatial pattern.

This will be modified accordingly.

(Page 2, Line 33) "poor ability" suggest using the inability.

This will be modified accordingly.

(Page 3, Line 1) "reached contrasting conclusions depending on the region or spatial scale, or on the variables and models considered." Given this, basing any conclusions regarding the role of internal variability on the fact that the models do not match the reconstructions seems problematic given that only two model grid points are being analyzed. A larger spatial scale might provide more confidence here, however the results using the CESM ensemble do provide strong evidence for the role of internal variability.

Actually we use more than 2 grid points, as shown in Fig. 1 of the submitted manuscript and discussed in Section 3.2. Furthermore, we have done the same analysis with modified study areas (including larger ones) and this does not make any substantial change. This will now be specified in Section 3.2:

Lake Challa and Lake Naivasha on the one hand and Lake Masoko and Lake Malawi on the other have a very similar climatology and seasonal cycle in the recent period, both in models and observations (Fig. 2). We thus consider a spatial domain which includes the first two lakes ( $0.2^{\circ}$  N to  $4.8^{\circ}$  S and  $34.2^{\circ}$  E to  $40.2^{\circ}$  E, referred to as the Challa/Naivasha region) and a second one which includes the last two ( $7.2^{\circ}$  S to  $12.2^{\circ}$  S and  $31^{\circ}$  E to  $37^{\circ}$  E, referred to as the Masoko/Malawi region; Fig. 1). Note that shifting or changing the size of these two regions to some extent does not generate substantially different results.

CPD
(Page 3, Line 20) It is not clear how the annual means follow from the long-term changes. Maybe save the discussion regarding the use of annual means portion for later.

Lake level depends, amongst other factors, on rainfall throughout the year. We thus hypothesize that the record is not seasonally biased, which is why we annually average the results of the climate models before smoothing them in order to match the temporal variability of the reconstructions.

(Page 3, Line 22) Is there any reason to suggest that models should capture the reconstructed changes? Perhaps change to analyze GCM simulations of long-term change relative to reconstructions over the last millennium. I do not think this study is truly aimed at investigating GCM performance.

We agree with the referee. This will be modified accordingly.

(Page 4) An aside, but I really appreciate the detail that you have gone into with regards to the set up of the models and the slight differences.

Thanks a lot for this comment.

Section 2.2. I think it is important to note how comparable each of these records are. Are there reasons to expect systematic differences given that they each are reconstructing different things? How might this impact the interpretation of the results and conclusions?

Cf. Our response to the referee's comment on section 4.1: The reconstructions are based on different proxies that have a different relationship with the target climate variable. However, as mentioned in Section 2.2, they are all appropriately sensitive to hydroclimate variation and can thus be qualitatively compared. We propose to add this sentence at the end of Section 2.2 to add clarity:

Although these records are derived from different proxies, their time series can all be qualitatively viewed as smoothed versions of these sites' local moisture-balance CPD
history, and should thus be related to common signals in their region's hydroclimate history.

Figure 2) On the x-axis Fev should be Feb, the labels overlap so potentially make text smaller.

This will be updated.

(Page 7, Line 15) To my eye CESM1 doesn't look better than the other models, BCC is arguably more realistic, maybe just remove the second part of that sentence.

We agree, the second part of the sentence will be removed.

(Page 8, Line 1) Put a space in "Lake Naivashais"

This will be done.

(Page 8, Line 4) I am not sure the split in how realistic the models are is that clear. When looking at CCSM4, CESM1 and BCC-CSM1-1, IPSL looks just as reasonable to me. Maybe be a bit more general.

This sentence actually refers to the average of all months ("Mean" column in Fig. 2). And although IPSL is not far, only the models CCSM4, CESM1 and BCC-CSM1-1 are in the range of the observations. We propose to rephrase to be clearer about what we are talking about:

For mean monthly precipitation throughout the year at Lake Challa (Fig. 2b), three models are within the range of the observations: CCSM4, CESM1 and BCC-CSM1-1.

(Page 8, Line 13) I think this is a bit of an oversimplification in the second part of this sentence as the timing is also off (not just the magnitude).

We agree and will mention that the timing can also be problematic.

(Page 8, Line 22) "Since Lake Challa and Lake Naivasha on the one hand and Lake Masoko and Lake Malawi on the other have a very similar climatology and seasonal cy-
cle both in models and observations (Fig. 2)". We are interested in long term changes, nothing that can really be done but the reconstructions for each lake look very different suggesting that on long timescales things might not be expected to be similar.

Yes, this is right. We show in Fig. 2 of the submitted manuscript that Lake Naivasha and Lake Challa on the one hand and Lake Masoko and Lake Malawi on the other hand are strongly linked, based on available recent observations and model results. This is confirmed in Section 3.2 using model results over the last millennium, but this cannot be confirmed using proxy-based reconstructions.

We propose to mention in the sentence cited by the referee that it only refers to the recent period:

Lake Challa and Lake Naivasha on the one hand and Lake Masoko and Lake Malawi on the other have a very similar climatology and seasonal cycle in the recent period, both in models and observations (Fig. 2).

(Page 8, Line 24) "However, using the larger grid boxes raises the issue of whether the proxy-based reconstructions employed to assess model performance through the last millennium are representative for these larger spatial domains". Be more exact here, don't use assess model performance. That's not really what's being done here.

Following the referee's comment, we propose to change the sentence by:

However, using larger grid boxes raises the issue of whether the proxy-based reconstructions that are compared to model results through the last millennium are representative for these larger spatial domains.

(Page 9, Line 12) "Regarding annual mean absolute values". What do you mean by this?

The term "absolute" means that we are talking about the true values, and not about anomalies. This will be specified in the manuscript.

CPD
(Page 10, Line 34) "Nevertheless, with values equal to 0.27 and 0.16 the significant correlation coefficients between, respectively." Are these spatial correlations? This section is a bit confusing; suggest revisiting with an eye towards clarifying the language.

We agree with the referee. To clarify the section, we propose to change:

"Out of the six GCMs, only CESM1 and MPI-ESM-P correctly simulate the spatial pattern of observed Challa/Naivasha rainfall (Fig. 4a). Nevertheless, with values equal to 0.27 and 0.16 the significant correlation coefficients between, respectively observed and simulated rainfall-SST correlation maps remain relatively low for these two models (Table S2)."

**by**

Out of the six GCMs, only CESM1 and MPI-ESM-P seem to correctly simulate the spatial pattern of correlations between Challa/Naivasha rainfall and SSTs (Fig. 4a). However, the match with observations is far from perfect as shown by the relatively low correlation coefficients between simulated and observed rainfall-SST correlation maps, with values equal to 0.27 and 0.16, respectively (Table S2).

(Page 12, Line 19) "theright", put space.

This will be done.

(Page 12, Line 27) "Making abstraction of chronological uncertainties, the Challa and Naivasha proxy records show discrepancy during the first four centuries of the last millennium". This sentence is confusing.

Cf. above, it will be replaced by:

Notwithstanding chronological uncertainties, the Challa and Naivasha proxy records display clear differences during the first four centuries of the last millennium.

(Page 12, Line 30) "humidity", suggest changing to wetting.
We will remove "in humidity".

(Page 13, Line 7) "Besides", suggest removing.

This will be done.

(Page 14, Line 2) "thus leads to larger river runoff towards the lakes". I am not sure that I understand this as the models do not have these actual lakes.

This is right, "towards the lakes" will be removed.

Page 14, Line 5) "This implies that most changes of P-E over time are due to changes in gprecipitation". Is this necessarily true on longer timescales?

One could indeed expect increasing importance of temperature and thus of evaporation when averaging the results over longer timescales. However, this is not the case in the models (Fig. 2 of this document). Using a loess filter with smoothing window of 100 years, the differences between the standard deviation of precipitation and evaporation drop by a factor of about 10, but the relative amplitudes remain approximately the same to what is shown using annual results, meaning that precipitation still dominates on evaporation.

**(Page 15, Line 8) Why not a more typical and interpretable standardization using the standard deviation?**

The only reason is aesthetic, dividing by the maximum allows limiting the amplitude of the time series between -1 and 1. In any case, using the standard deviation does not change dramatically the results, as you can see in Fig. 3 of this document.

(Page 15, Line 9) How does this or does this not match the temporal resolution of those reconstructions. I suppose I am just interested in a more thorough justification for this choice of smoothing.

The exact resolution of each reconstruction is variable through time because of the non-linear relationship between sediment depth and age in all four time series. There-

CPD
fore, here we just estimate it at a few decades on average. We chose to smooth the model results using a window of 100 years, which appears to be a good value to make model variability qualitatively similar to that in the reconstruction. Note that we tested several window widths, but this did not change the results significantly. This will be specified in the revised version:

Our choice of 100-year smoothing window is partly subjective, since the resolution of the proxy-based reconstructions varies through time due to a non-linear relationship between sediment depth and age. However, using other window widths for this smoothing does not lead to major changes in the results.

(Page 15, Line 11) By eye in Figure 7 the model variability looks to be about the same magnitude as the reconstructions.

We consider that the large (multi-)centennial fluctuations visible in the reconstructions are not present in the model results. Indeed, the reconstructions can be positive or negative for several consecutive centuries, while the model curves usually only for a couple of decades (unless one takes into account the memory by using an autore-gressive model). Still, we propose to change "much weaker fluctuations" by "weaker fluctuations" to better describe what can be seen on Fig. 7 of the manuscript.

(Page 16, Line 12) "confronted", should be compared.

This will be updated.

(Page 17, Line 10) This is an important sentence. At least remove "that" to make it clear, however, restructuring the sentence would probably be good.

We agree with the reviewer. We propose to modify as follows:

"The fact that all the model results appear different raises the question whether external forcing has any impact on the simulated hydroclimate, forcing that is comparable between models (see Section 2.1) and is thus expected to put a comparable imprint on all time series." CPD
by

Simulations with different GCMs take into account comparable climate forcing (see Section 2.1), which is thus expected to put a comparable imprint on all time series. However, all model results seem different, which raises the question whether external forcing has any impact on the simulated hydroclimate.

(Page 17, Line 22) "In this regard, it is of interest to note that for the one model for which multiple ensemble members were available (CESM1), there is also no correlation between the different ensemble members that differ only from slightly different air temperature at the start of the experiments (Otto-Bliesner et al., 2015)". This is very important and gets lost a bit as cast.

We agree with the referee that this is an important sentence. In order to better highlight it, we propose to put a paragraph break just before.

(Section 5.2) I feel like this section would be clearer with the unfiltered pre-industrial control run teleconnections also shown. It becomes a bit confusing with the pre-industrial portion of the last millennium, pre-industrial control runs, and historical simulations all being compared simultaneously.

We think that adding even more material could actually bring confusion in this Section that is indeed already quite dense. Moreover, it contains only one reference to the figure showing the unfiltered pre-industrial control run teleconnections (p.20 I.9), so we prefer to keep this figure in the supplementary material (Fig. S3).

The historical simulations are not shown in Section 5.2, as is mentioned in its first paragraph. The 3 figures included show the annual teleconnections according to the 'past1000' simulations (period 850-1850; Fig.10), and the smoothed teleconnections according to the 'PI control' simulations (Fig. 11) and to 'past1000' simulations (Fig. 12).

Nevertheless, this section will be modified in the revised version to add clarity.
Conclusion) This ends on a weak note, I might suggest finishing the paper with the last sentence of the previous paragraph.

Actually, we think that this does not end on a weak note, but on a caution that can be important for future studies dealing with hydroclimate as simulated by GCMs. Furthermore, the last sentence of the previous paragraph only relates to the teleconnections, while the last paragraph is more general. We would thus prefer to keep it the way it is.

---

## Author Comment (AC3) · 10 May 2016

- In blue: referees' comments

- In black: our answers

- *In black italic: what we propose to add in the text*

This paper deals with a very important subject: Comparison of the simulated and reconstructed climate. Working out similarities and differences is key for a better understanding of the climate drivers and their quantification.

The authors have chosen four fairly high-resolution climate curves from East Africa

which they compare with model results. Interestingly, some of the reconstructed climate curves differ markedly. In Figure 5 the Naivasha and Masoko lakes show a dry Medieval Warm Period (MWP) / Medieval Climate Anomaly. In contrast, Challa appears to be more humid, even though the record starts slightly later and the beginning is unclear. In the Lake Malawi climate curve the time 1000-1300 AD is absent, therefore it is unclear if the MWP was dry or wet here. I suggest you add information from Johnson et al. 2004.

According to those authors: "Diatom productivity was high during the Little Ice Age (LIA) and relatively low around 1 kyr, the time of the Medieval Warm Period (MWP)". The low diatom productivity during the MWP may be linked to low river discharge, i.e. drought conditions. During this time the rivers may have supplied lower amounts of dissolved silica to the lake. During the wetter Dark Ages Cold Period and Little Ice Age, chemical weathering of bedrock intensified and increased the BSi concentrations and diatom productivity in the lake. http://link.springer.com/chapter/10.1007

Thank you for your comment. The reconstruction of Johnson et al. (2004) is certainly interesting, but we prefer not to discuss it in the revised version, since it would add complexity to our manuscript (which is already long as underlined by the reviewers) without adding much extra value for the model-data comparison. Indeed, our goal here is not to extensively review the available proxy-based reconstructions of hydroclimate, but to compare model simulations with available reconstructions that have a sufficient temporal resolution to make the comparison meaningful. Nevertheless, we have now mentioned in the Section 2.2 of the manuscript (Proxy-based hydroclimate reconstructions) that comparing to additional records would be an interesting follow up to the present work (see sentence below).

I would like to draw your attention to an ongoing project in which I am mapping the climate characteristics of the MWP on a global scale, based on the large number of published case studies. The interactive online map is freely accessible here: http://t1p.de/mwp

In East Africa you see a large number of yellow points that represent studies which reported drought/arid conditions for the MWP time. When you click on the respective dot, key information from the paper appears, including a link to the key climate curve. Arid conditions seem to be the general pattern that existed 1000-1300 AD in East Africa. The arid MWP belt appears to continue northwards along the coast of the Arabian Sea, including Ethiopia, Yemen, Oman, Pakistan and coastal northwestern India. There, the MWP climate regime seems to change. Southern and eastern India and the Bay of Bengal appear to be humid during the MWP. Mapping is still ongoing and many more studies will have to be integrated. It is also clear that in detail things are more complex. Nevertheless, I think it would be important to initially compare the models to these general, high-level patterns.

Thank you very much for the information. The map is really interesting, showing large-scale features whilst taking into account local patterns. In our manuscript, we have chosen to use the compilation of hydroclimate proxy-based reconstructions for our study region, as done in Tierney et al. (2013). Although your map contains some East African hydroclimate reconstructions that may be interesting to compare with model results, it appears that the number and type of the reconstructions used in our study is not critical to reach our conclusions, given that we suggest that simulated hydroclimate is mainly driven by internal variability in this region. Hence, we do not expect to find much agreement between model results and hydroclimate reconstructions over the last millennium, and, if any was found, it would be coincidental only. However, a comparison with additional records would certainly be interesting to assess the robustness of the changes inferred from the records selected here and to refine the spatial structure of those changes. This is now specified as a possible future perspective of our work, in Section 2.2 of the manuscript (Proxy-based hydroclimate reconstructions):

*Several other proxy records exist describing East African hydroclimate variability during the last millennium (Verschuren, 2004; see also http://t1p.de/mwp). The majority of these mostly lake-based records do not possess sufficient age control to assess the*

*regional coherence of inferred (multi-) decadal and century-scale hydroclimate varia-
tion. Since our goal here is not to extensively review the strengths and weaknesses of
those individual reconstructions, we follow Tierney et al. (2013) to consider only the
handful of records which combine high temporal resolution with adequate age control.
However, a critical review of all available records could potentially refine the spatial
structure of documented hydroclimate changes and allow further assessment of the
robustness of broad-ranging climate-dynamic inferences.*

From your study and reference list I have gathered quite a few new publications that I
will add to the MWP map in due course. Thanks for that.

Thanks for the comment.

Concerning the forcing of pre-industrial climate change, I am not comfortable with mod-
els that gain their simulated climate variability mostly from internal variability. There
are clear MWP patterns and additional millennium cycles (e.g. Bond et al. 2001)
which point towards powerful external climate drivers. Numerous papers have high-
lighted the important role that solar activity changes play in the climate equation. I
want to encourage you to also run models and scenarios with a solar radiative forc-
ing higher than that assumed by the IPCC. If not for this paper, maybe in a future
one. The current RF proposed by the IPCC does not honour the great number of
studies which highlight the intense coupling of climate with solar activity changes:
http://chrono.qub.ac.uk/blaauw/cds.html

Indeed, many studies have emphasized solar forcing of climate change, based on com-
parison between proxy reconstructions of different climate variables (temperature, rain-
fall, etc.) and reconstructed solar activity variations. However, the changes in total solar
irradiance are small. To induce a significant impact on the climate system, feedback
mechanisms are required to amplify these initial changes, but these mechanisms are
not well known. Furthermore, previous studies have had trouble to formally detect the
influence of solar irradiance on climate of the past millennium and to determine whether

simulations with low or moderate solar forcing are more realistic (e.g., Schurer et al., 2014; Jungclaus et al. 2010, PAGES2k-PMIP 2015). By contrast, simulations driven by very large solar forcing are not compatible with many available reconstructions. Running new model experiments with enhanced solar forcing could provide insights into this controversial topic, but unfortunately, it will not be possible to include this in the present study.

Here, we have not run any simulations ourselves. Making new simulations covering the last millennium with these GCMs requires a tremendous amount of computer time, in addition to technical expertise adapted to each model. We have thus used the publicly available experiments that were performed by different modeling groups. All those simulations have been performed following the well-defined frameworks (changes in forcing, periods covered by the simulations, etc) of the projects PMIP3 (for the past1000 simulations, from 850 AD to 1850 AD, Otto-Bliesner et al., 2009) and CMIP5 (for the historical simulations, from 1850 AD to present, Taylor et al., 2012). Making new coordinated model experiments with enhanced solar forcing would thus require a joined and long-term effort, and although potentially very interesting, is out of scope of this study.

**References**

Brown, E. T. and Johnson, T. C.: Coherence between tropical East African and South American records of the Little Ice Age, Geochemistry, Geophysics, Geosystems, 6, 1–11, doi:10.1029/2005GC000959, 2005.

Johnson, T.C., E.T. Brown, and J. McManus. (2004) Diatom productivity in Northern Lake Malawi during the past 25,000 years: The intertropical convergence zone and comparisons to other high resolution paleoclimate records from East Africa. In: R.W. Battarbee, F. Gasse, and C.E. Stickley eds. Past climate variability through Europe and Africa. Kluwer Academic.

Jungclaus, J. H., Lorenz, S. J., Timmreck, C., Reick, C. H., Brovkin, V., Six, K.,

Segschneider, J., Giorgetta, M. A., Crowley, T. J., Pongratz, J., Krivova, N. A., Vieira, L. E., Solanki, S. K., Klocke, D., Botzet, M., Esch, M., Gayler, V., Haak, H., Raddatz, T. J., Roeckner, E., Schnur, R., Widmann, H., Claussen, M., Stevens, B., and Marotzke, J.: Climate and carbon-cycle variability over the last millennium, Clim. Past, 6, 723–737, doi:10.5194/cp-6-723-2010, 2010.

Otto-Bliesner, B. L., Joussaume, S., Braconnot, P., Harrison, S. P., and Abe-Ouchi, A.: Modeling and Data Syntheses of Past Climates: Paleoclimate Modelling Intercomparison Project Phase II Workshop, Eos, Transactions American Geophysical Union, 90, 2009.

PAGES 2k-PMIP3 group: Continental-scale temperature variability in PMIP3 simulations and PAGES 2k regional temperature reconstructions over the past millennium, Climate of the Past, 11, 1673–1699, doi:10.5194/cp-11-1673-2015, 2015.

Schurer, A. P., Tett, S. F., and Hegerl, G. C.: Small influence of solar variability on climate over the past millennium, Nature Geoscience, 7, 104–108, doi:10.1038/NGEO2040, 2014.

Taylor, K. E., Stouffer, R. J., and Meehl, G. a.: An Overview of CMIP5 and the Experiment Design, Bulletin of the American Meteorological Society, 93, 485–498, doi:10.1175/BAMS-D-11-00094.1, http://journals.ametsoc.org/doi/abs/10.1175/BAMS-D-11-00094.1, 2012.

Tierney, J. E., Smerdon, J. E., Anchukaitis, K. J., and Seager, R.: Multidecadal variability in East African hydroclimate controlled by the Indian Ocean., Nature, 493, 389–92, doi:10.1038/nature11785, http://www.ncbi.nlm.nih.gov/pubmed/23325220, 2013.

Verschuren, D.: Decadal and century-scale climate variability in tropical Africa during the past 2000 years, in: Past Climate Variability through Europe and Africa, edited by Battarbee, R. W., Gasse, F., and Stickley, C. E., chap. 8, pp. 139–158, Dordrecht, The Netherlands, springer edn., 2004.